# Quantitative prediction of grain boundary thermal conductivities from local atomic environments

Susumu Fujii [1,2,3✉], Tatsuya Yokoi[3,4], Craig A. J. Fisher [1], Hiroki Moriwake[1,2] & Masato Yoshiya [1,3,5✉]

Quantifying the dependence of thermal conductivity on grain boundary (GB) structure is critical for controlling nanoscale thermal transport in many technologically important materials. A major obstacle to determining such a relationship is the lack of a robust and physically intuitive structure descriptor capable of distinguishing between disparate GB structures. We demonstrate that a microscopic structure metric, the local distortion factor, correlates well with atomically decomposed thermal conductivities obtained from perturbed molecular dynamics for a wide variety of MgO GBs. Based on this correlation, a model for accurately predicting thermal conductivity of GBs is constructed using machine learning techniques. The model reveals that small distortions to local atomic environments are sufficient to reduce overall thermal conductivity dramatically. The method developed should enable more precise design of next-generation thermal materials as it allows GB structures exhibiting the desired thermal transport behaviour to be identified with small computational overhead.

[1] Nanostructures Research Laboratory, Japan Fine Ceramics Center, 2-4-1 Mutsuno, Atsuta, Nagoya 456-8587, Japan. [2] Center for Materials Research by Information Integration, National Institute for Materials Science, 1-2-1 Sengen, Tsukuba, Ibaraki 305-0047, Japan. [3] Department of Adaptive Machine Systems, Osaka University, 2-1 Yamadaoka, Suita, Osaka 565-0871, Japan. [4] Department of Materials Physics, Nagoya University, Furo-chou, Chikusa, Nagoya 464-8603, Japan. [5] Division of Materials and Manufacturing Science, Osaka University, 2-1 Yamadaoka, Suita, Osaka 565-0871, Japan. ✉email: susumu_fujii@jfcc.or.jp; yoshiya@ams.eng.osaka-u.ac.jp

Thermal conductivity is a fundamental property of a material and crucial for many technological applications, e.g., thermoelectrics[1–3], thermal barrier coatings[4,5], high-power devices[6,7] and microelectronics[8,9]. Recent studies have shown that nanocrystalline materials, which have large grain boundary (GB) populations, exhibit extremely low lattice thermal conductivities[1,5,10,11], even when the bulk form is thermally conductive, e.g., elemental silicon[12,13]. This dramatic reduction in lattice thermal conductivity is commonly attributed to shortening of the phonon mean free path (MFP), with the assumption that it is of the same order as the average grain size[5,12,14,15]. Although this first-order approximation has informed most attempts to control thermal conductivity, e.g., by tailoring grain size distributions[16], it does not take into account the impact of individual GBs and their different atomistic structures, and recent experimental studies have indicated that the amount of thermal conductivity reduction varies considerably depending on the structure of a particular GB[17–19]. For example, Tai et al[18]. measured the thermal resistances of three twist $Al_2O_3$ GBs and found that they vary by a factor of three. Quantitatively determining the relationship between GB structure and thermal conductivity is thus desirable for designing thermally functional materials at the nano-scale.

Many computational studies have been performed over the past two decades using non-equilibrium molecular dynamics (MD) to examine thermal conductivities of individual GBs[20–24]. Although the results revealed that thermal conductivity varies with misorientation angle and GB energy, the underlying physical mechanism responsible for this has not been elucidated in terms of the GB structures themselves. To help remedy this, we recently calculated thermal conductivities of 81 MgO symmetric tilt GBs (STGBs), and found that GB excess volume, which stems from reduced atomic coordination and non-optimal bond lengths at the GB core (the characteristic structure pattern centred on the GB plane), is strongly correlated with thermal conductivity[25]. We identified three different correlations depending on the type of GB, with low thermal conductivities occurring in the vicinity of the most open structures. The results provided further evidence that thermal conductivity can vary significantly depending on the type of GB and its atomic structure.

An analysis based on excess volume alone, however, is insufficient for explaining structure-property relationships over high-dimensional space, e.g., general GBs in polycrystals, because a given excess volume is not necessarily unique to a particular GB structure. This is because excess volume is a measure of the non-optimum packing of atoms at a GB but contains no other information about how the GB structure differs from that in the crystal bulk or to other GBs; consequently two GBs can have the same excess volume but exhibit very different thermal conductivity behaviour because of differences in atom configurations and bonding[26–30]. General GBs consist of complex mixtures of simpler high-symmetry (planar) GBs, and are even harder to analyse because of the enormous number of degrees of freedom involved. This problem is exacerbated when the effect of intrinsic defects or impurity atoms is included. A brute force method, e.g., MD simulation, can enable a specific thermal conductivity to be assigned to a specific GB core structure so that the dependence of thermal conductivity on GB misorientation and composition can be examined systematically, but even using computationally inexpensive empirical potential models it would take an inordinately long time to generate sufficient data for a wide variety of GB forms. Thus a more efficient and computationally tractable method is needed if meaningful progress is to be made.

A promising method for handling large numbers of different atomic configurations is the use of structure descriptors developed in the burgeoning field of materials informatics[31–35]. These descriptors contain information sufficient to define uniquely a particular atom arrangement, and act as fingerprints distinguishing different atomistic structures. Recent studies have used such descriptors in the context of machine learning (ML) to enhance our understanding of GB structure-property relationships[36–38]. A prime example is the study of Rosenbrock et al.[38]; using the smooth overlap of atomic positions (SOAP) descriptor[39,40] and a supervised ML technique, they identified a set of building blocks (or representative local atomic environments, LAEs) from which GBs of metallic Ni are constructed, and determined which LAEs strongly influence GB energies and mobilities. In related work[41] they reviewed various models used to analyse GB structures (in particular comparing the utility of the local environment representation to that of the structural unit model in the analysis of 126 Ni STGBs), and showed that the former is in many respects superior to the others, most notably because it provides a smoothly varying function.

In this report we describe our search for a suitable SOAP-based microscopic metric that correlates with GB thermal conductivity and can be used to identify relationships between GB structure and thermal conductivity. To ensure the rigour of the relationship identified, a wide range of GBs are included in the analysis, viz., symmetric tilt, twist, twin and asymmetric tilt GBs stable at standard pressure, and symmetric tilt GBs stable at higher pressure. MgO is chosen as a model material because of its simple structure and long history of experimental and theoretical work. The most appropriate microscopic quantity that we identify, which we refer to as the local distortion factor, LDF, measures deviations in the local structural environment of an atom near a GB from that of an identical atom in the crystal bulk, and correlates well with atomically decomposed thermal conductivities perpendicular to the GB extracted from perturbed MD simulations. We then construct a prediction model using multiple linear regression with input variables based on hierarchical clustering of LAEs, and demonstrate that the thermal conductivity of a GB can be predicted with high accuracy using this model. Analysing the results in terms of LDFs reveals that even a small amount of structural distortion at the GB is sufficient to suppress thermal conductivity strongly. We expect that extension of this ML-based technique to other materials should greatly enhance our understanding of GB behaviour, thereby enabling materials to be tailored to exhibit the desired thermal properties, especially once suitable nano-scale engineering techniques have been developed.

## Results

**Effective thermal conductivities**. In addition to low-angle and high-angle STGBs reported previously[25], in this study we calculated effective thermal conductivities across GB planes of standard-pressure twist, asymmetric tilt and high-pressure tilt GBs of MgO to obtain a more comprehensive understanding of the relationship between GB structure and thermal conductivity. Detailed lists of all GB models used in this study are provided in Supplementary Tables 1–9, with some relevant properties summarised in Supplementary Figs. 1–3, and explanatory notes included as Supplementary Notes 1 and 2. The combined results are plotted in Fig. 1a against excess volume per unit area of each GB, with representative GB structures shown in Fig. 1b–h. For the STGBs under standard pressure, the thermal conductivities exhibit three different correlations with excess volume depending on the GB type: low-angle GBs with (I) dense and (I′) open dislocation core structures, and (II) high-angle GBs. In Fig. 1a, thermal conductivities of high-angle high-pressure STGBs also fall on correlation line II (solid black line), whereas their excess volumes are smaller than those of standard-pressure STGBs with the same misorientation because of their denser GB core

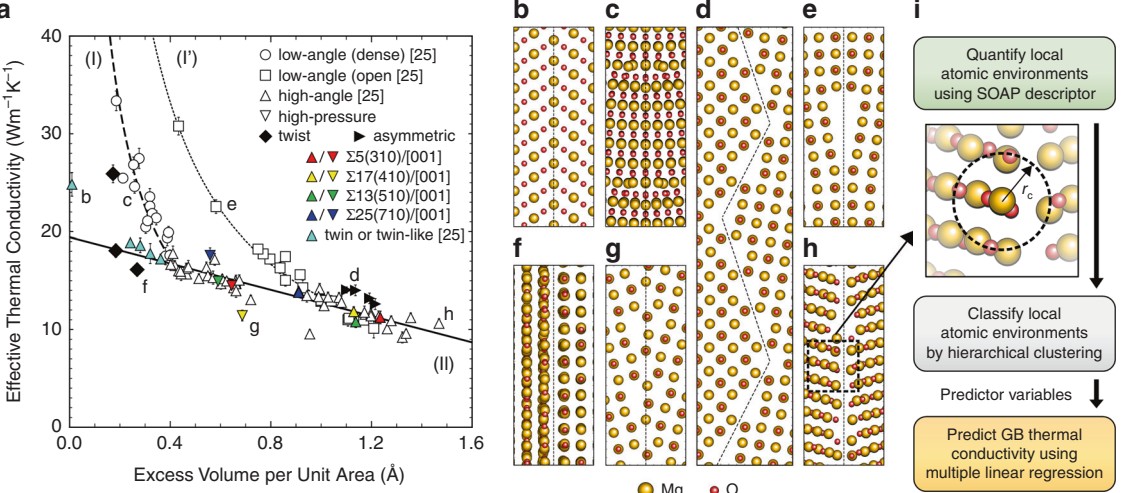

**Fig. 1 Overview of GB thermal conductivities and structures. a** Effective thermal conductivities across standard-pressure tilt and twist GBs, twin (and twin-like) boundaries, and high-pressure tilt GBs as a function of excess volume. Data for standard-pressure tilt GBs and the correlations indicated by solid, dashed and dotted lines are from Fujii et al.[25] (I) low-angle tilt GBs with dense dislocation core structures; (I′) low-angle tilt GBs with open dislocation core structures; and (II) high-angle tilt GBs. Error bars indicate standard deviations in thermal conductivity calculated using perturbations of different magnitudes. **b–h** Example structures of different types of GB: **b** twin, **c** low-angle GB with dense dislocation cores, **d** asymmetric tilt GB, **e** low-angle GB with open dislocation cores, **f** twist GB, **g** high-pressure high-angle GB and **h** standard-pressure high-angle GB. **i** Method for predicting GB thermal conductivities based on local atomic environments. $r_c$ is the cutoff radius of the SOAP descriptor.

structures. In contrast, thermal conductivities of low-angle high-pressure STGBs deviate from these trends, lying between lines I and I′ because of the intermediate densities of their dislocation structures. Thermal conductivities of asymmetric Σ5 [001] tilt GBs lie only slightly above the correlation II line, probably because their GB core structures are similar to those of high-angle GBs; the asymmetric boundaries in this study are mainly composed of (310) and (210) facets similar to those in the corresponding symmetric boundaries, although different kinds of atomic structures are formed at the facet junctions.

The three twist GBs examined also show similar behaviour to the dense low-angle STGBs; the thermal conductivity is high for low excess volumes, and initially decreases rapidly with increasing excess volume, but the rate of decrease diminishes once the dislocations begin to overlap. The most pertinent difference between twist and tilt GBs in this case, however, is that the excess volumes of twist GBs are much smaller than those of STGBs because of their denser structures. GBs with very high symmetry and thus high number density, viz., the Σ3(111) twin boundary and GBs with LAEs similar to it (labelled twin-like in Fig. 1a), appear to fall on a fourth correlation line, one flatter than correlations I or I′ (see Supplementary Fig. 1 for their structures). The results indicate that the thermal conductivities of high-pressure tilt, asymmetric tilt and twist GBs are governed by the same mechanism as for STGBs, and that the macroscopic metric, i.e., GB excess volume, is inadequate as a parameter for accurately predicting thermal conductivities of various types of GB structures. As explained below, we overcome this problem by quantifying LAEs in the vicinity of GBs using the SOAP descriptor to generate input data for ML techniques. A schematic of the method is shown in Fig. 1i.

**Quantifying local distortions**. The mechanism by which thermal conductivity is reduced at GBs is expected to be related to local structural distortions because long-range thermal transport occurs by phonons, which are the collective motion of atoms in a periodic lattice, and any disturbance to this motion results in enhanced phonon scattering, as evidenced by numerous experimental and theoretical studies[1,3,5,10–25]. To quantify these

structural distortions, we defined a (non-normalised) dissimilarity metric that measures the difference in LAE between an atom at a GB and an atom in the crystal bulk, which we refer to as the local distortion factor, LDF using the SOAP descriptor (see Methods for details). We calculated LDFs of all atoms in GB structure models for a wide variety of different GB types, viz., 80 standard-pressure STGBs (about six different rotation axes), a twin[25], four high-pressure [001] STGBs, three (001) twist GBs and four asymmetric [001] tilt GBs. Figure 2a shows a plot of the LDFs in each model classified by GB rotation axis in order of increasing tilt or twist angle. The LDFs span a wide range, from 0 to 3000, with atoms at open GBs tending to have high values and those at relatively dense GBs to have low values.

To quantify how LDFs vary with bond elongation, we also calculated those of atoms in uniformly expanded, defect-free MgO single crystals, and the results are plotted in Fig. 2b. This plot shows that when an MgO crystal is expanded isotropically, LDFs (those of cations and anions are equivalent in this case because of its rock-salt structure) increase smoothly and reach a value equivalent to the maximum LDF in the GB models for a lattice constant elongation of ~9.5% and volume expansion of ~31.2%. Unlike the atoms in the perfect crystal, atoms at GBs are not subjected to as large increases in local volume or bond lengths, but instead experience non-uniform (anisotropic) strain to their bonds and/or changes in coordination environment. Although LDFs by themselves do not indicate whether strain or coordination environment has the stronger effect, separate analysis showed that both of them are important, with contributions of similar magnitude in many cases. For example, the average and standard deviation of LDFs of atoms with first-nearest neighbour coordination deficiencies of 0, 1 and 2 are $456.6 \pm 392.4$, $1270.0 \pm 608.4$ and $1483.5 \pm 580.0$, respectively. The LDF values increase with increasing under-coordination but also have high standard deviations because of large variations in bond strain about atoms with different LAEs.

**Clustering analysis of LAEs**. To classify the structural environments of atoms at the cores of different GBs into groups suitable for constructing our ML model, we first used the complete-linkage

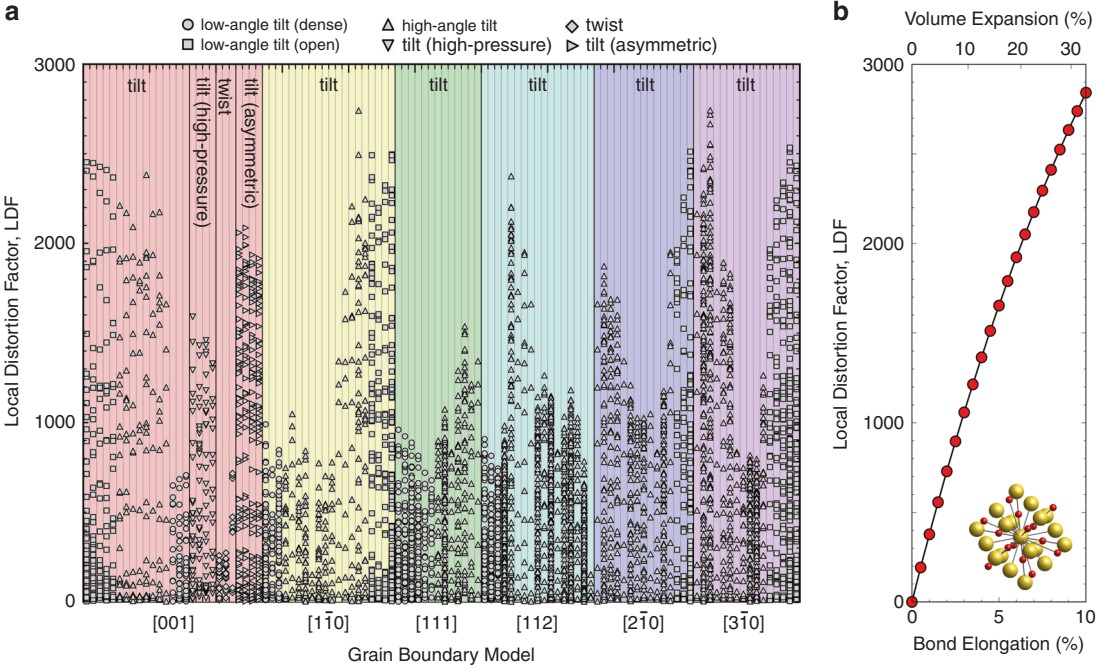

**Fig. 2 Local distortion factors, LDFs, obtained using the SOAP descriptor. a** LDFs of all LAEs in tilt and twist GB models in order of increasing tilt or twist angle for each class of rotation axis. One vertical column of LDFs corresponds to one GB model. **b** LDFs of atoms in an isotropically expanded unit cell as a function of bond elongation and volume expansion.

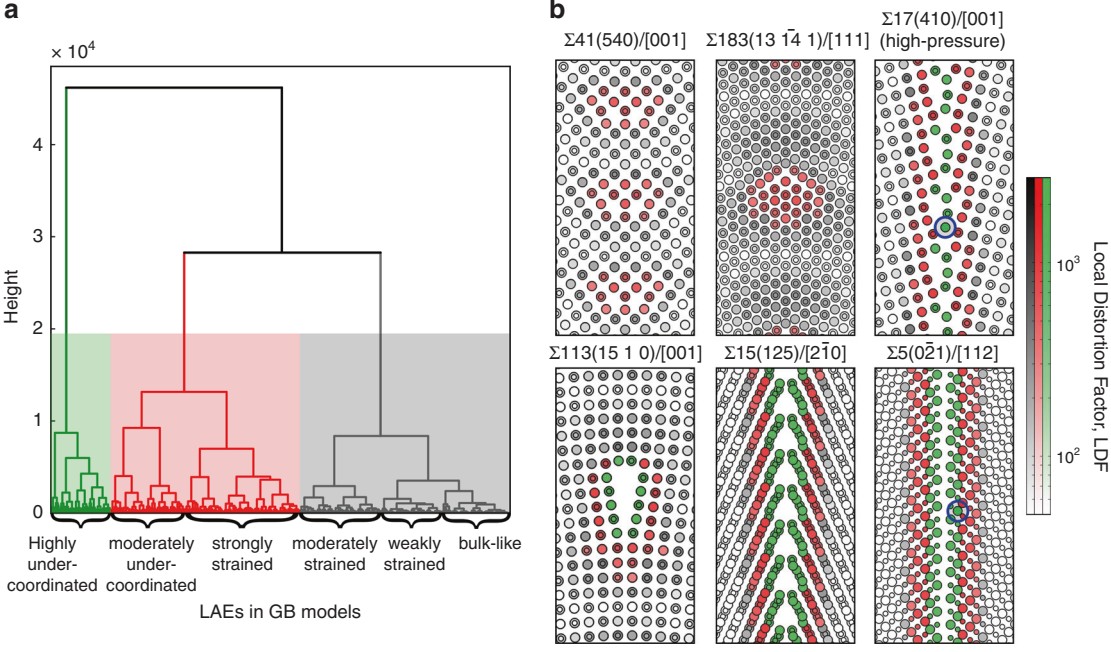

**Fig. 3 Hierarchical clustering of GB LAEs. a** Hierarchical relationship between LAEs depicted in dendrogram form. The different regions represent three general groups of LAEs: (green) highly under-coordinated (bond-ruptured); (red) moderately under-coordinated or strongly strained; and (grey) moderately strained, weakly strained or bulk-like. **b** Representative distributions of the LAE groups and LDFs at six STGBs. A log scale is used to make it easier to distinguish changes in LDFs within LAE groups.

method to identify LAEs in each GB model based on the dissimilarity metric $d$ between each pair of atoms, before applying Ward's hierarchical clustering method[42] to the complete set of LAEs generated (see Methods for details). Figure 3 shows a dendrogram of the different classes of LAEs identified, together with representative STGBs to illustrate how they are distributed around GBs. The dendrogram in Fig. 3a shows three supergroups of LAEs (indicated by different colour shading in the figure) that are classified into six groups whose members consist of unstrained (bulk-like) atoms, weakly strained atoms, moderately strained atoms, strongly strained atoms, moderately under-coordinated atoms and highly under-coordinated (bond-ruptured) atoms. The averages and standard deviations of LDFs in these six groups are $70.0 \pm 66.1$, $138.7 \pm 42.9$, $316.8 \pm 84.8$, $609.8 \pm 140.7$, $1032.3 \pm 165.6$ and $1786.1 \pm 323.8$, respectively, reflecting the increasing amount of structural distortion (LDF distributions in each LAE group are

reported in Supplementary Fig. 3). For reference, from Fig. 2b the LDFs of weakly strained, moderately strained and strongly strained groups correspond to average bond elongations of roughly 0.4, 0.8 and 1.6%, respectively. The average amounts of first-nearest neighbour under-coordination in these groups are 0.00, 0.01, 0.03, 0.22, 0.37 and 0.94, respectively, suggesting that the effect of strongly strained atoms is of similar magnitude to that of slightly under-coordinated atoms.

Figure 3b shows GB structures coloured according to LAE group and LDF values for two low-angle STGBs with dense structures, a low-angle STGB with open structure, a high-pressure high-angle STGB and two standard-pressure high-angle GBs. These indicate that highly under-coordinated atoms occur at open GB core structures, whereas dense GB core structures consist of atoms in strongly strained environments, either at dense low-angle GBs or adjacent to under-coordinated atoms in high-angle STGBs. In dense low-angle GBs such as $\Sigma183(13\overline{14}1)/[111]$ and $\Sigma113(15\,1\,0)/$ [001], atoms between the dislocation cores have LAEs similar to bulk atoms. These results illustrate how hierarchical clustering of LAEs and LDFs captures information regarding the arrangement of atoms and degree of distortion at GBs in a physically interpretable manner.

LDF values quantify the local distortion relative to the ideal crystal bulk, but do not directly measure differences in LAEs between GBs. To better assess the range of LAEs present in different types of GBs, we thus also calculated $d$ values between all atoms in one GB model with those in another. This revealed that similar LAEs frequently occur in other GBs, with greater differences occurring for high-pressure and high-angle STGBs than for others. Specifically, the minimum $d$ value of any atom in the high-pressure STGBs, asymmetric tilt GBs and twist GBs were no greater than 211, 140 and 87, respectively (compared to maximum LDFs close to 3000); these values correspond to about 0.5%, 0.4% and 0.2% bond elongation, respectively, when considered in terms of a uniformly expanded MgO crystal (Fig. 2b). For example, the $d$ value for the two atoms indicated by blue circles in the high-pressure $\Sigma17(410)/$ [001] GB and standard-pressure $\Sigma5(0\overline{2}1)/[112]$ GB in Fig. 3b is only 58.2. In other words, the range of LAEs provided by a sufficiently large and diverse sample of GB structures (92 in our case) is expected to encompass those encountered in GBs with other misorientations, higher complexity or lower symmetry. This result is consistent with Priedman et al.'s observation that different GBs consist of similar structural building blocks or motifs[41]. Consequently, similar to Rosenbrock et al.'s[38] findings for GB energies and mobilities, the properties and behaviour of individual GBs of MgO can be expected to depend on the relative numbers of each type of LAE of which they are composed. Identifying correlations between the numbers and distributions of LAEs in a GB and its thermal conductivity, preferably in a physically meaningful way, should thus allow thermal conductivities of MgO GBs of arbitrary structure to be predicted quickly, accurately and reliably. In the following sections we demonstrate how hierarchical clustering can fulfil this purpose in the context of thermal transport and phonon dispersion, with interpretation facilitated by analysing LDFs.

**Thermal conduction at tilt grain boundaries**. To determine the dependence of microscopic thermal conduction on structural distortion in the vicinity of GB planes, we calculated atomic thermal conductivities perpendicular to GB planes at 300 K using perturbed MD simulations, and LDFs from the relaxed GB structures for each GB model.

Figure 4 compares plots of LDFs and atomic thermal conductivities of standard- and high-pressure $\Sigma25(710)/[001]$ and $\Sigma5(310)/[001]$ STGBs, together with the LAE classifications identified by hierarchical clustering. These plots reveal that,

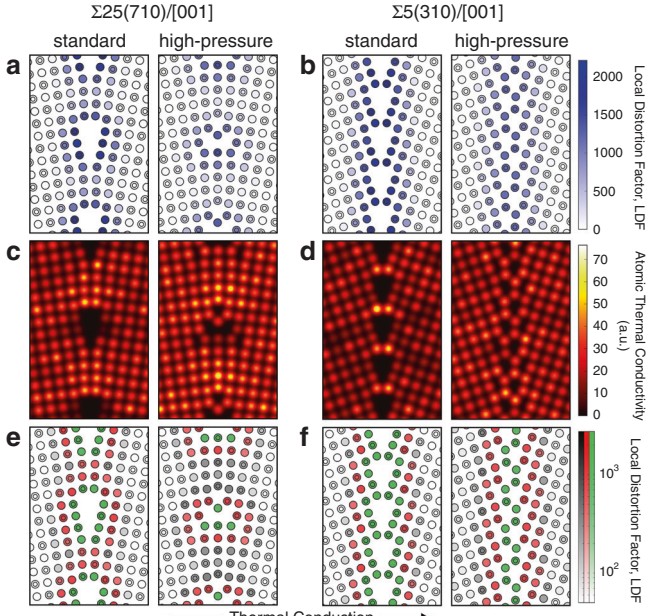

**Fig. 4 Atomic configurations and atomic thermal conductivities near GB planes of four STGBs. a**, **b** Local distortion factors, LDFs; **c**, **d** Gaussian-smeared atomic thermal conductivities; **e**, **f** Distributions of LAE groups classified from hierarchical clustering. A log scale is used to make it easier to distinguish changes in LDFs within LAE groups.

overall, there is strong negative correlation between LDF and atomic thermal conductivity in these two cases. One exception to this is the standard-pressure $\Sigma5(310)/[001]$ STGB, in which LDFs of the innermost atoms (Fig. 4b) are high and their atomic thermal conductivities (Fig. 4d) are the highest of all atoms in the GB structure. This inversion of the correlation is because the SOAP vector, and hence LDF, are non-directional, whereas there is a large anisotropy in the bond distances and hence components of atomic thermal conductivity of the $\Sigma5(310)/[001]$ GB, with single pairs of atoms across the GB plane acting like thermal conduction bottlenecks. Distances between atoms perpendicular to the GB plane are similar to those in the bulk, but much longer parallel to it in the $[1\overline{3}0]$ direction, resulting in a large LDF factor (maps of the components of atomic thermal conductivity perpendicular and parallel to the GB plane are compared in Supplementary Fig. 4 and Supplementary Note 3). Such bottle-necks generally only occur in high-angle STGBs, but in low densities dispersed between low-conductivity voids, so their effect on the overall thermal conductivity is small.

The greatest decrease in atomic thermal conduction occurs at the centres of dislocation cores, whereas thermal conduction is rapid via atoms in less disturbed (low LDF) regions even if on the GB plane (corresponding to light-coloured atoms in Fig. 4a, b). The core structures of the high-pressure GBs are denser than those of the standard-pressure GBs, making them more like low-angle GBs in which dislocations are arrayed in a regular pattern. This results in the wider regions of unruptured bonds on the GB planes seen in the right-hand images of Fig. 4a, c. This explains why the effective thermal conductivity of the low-angle high-pressure GB is higher than those of the standard-pressure GBs, falling between lines I and I' in Fig. 1a because of the intermediate atomic densities of its dislocations. Overall, the close correspon-dence between LDF and atomic thermal conductivity suggests that this metric makes a good descriptor for developing a model for predicting thermal conductivities in a wide variety of GB types of MgO.

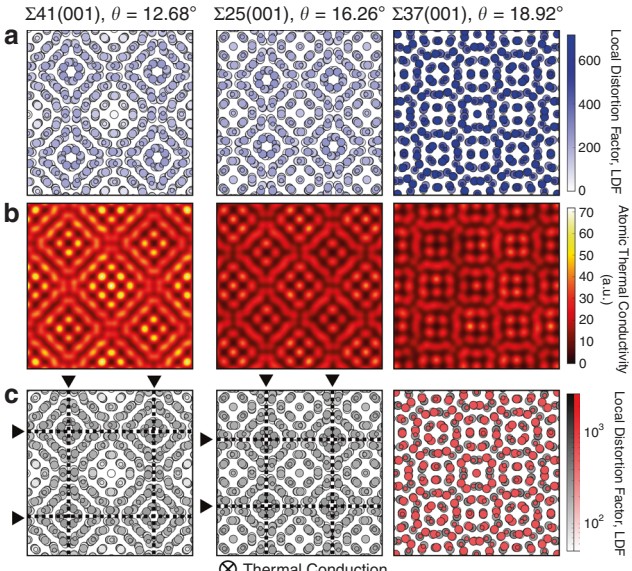

Σ41(001), θ = 12.68°    Σ25(001), θ = 16.26°    Σ37(001), θ = 18.92°

**Fig. 5 Atomic configurations and atomic thermal conductivities near GB planes of three twist GBs. a** Local distortion factors, LDFs; **b** Gaussian-smeared atomic thermal conductivities; **c** Distributions of LAE groups classified using hierarchical clustering. Dislocation lines are shown as dashed lines in **c**. A log scale is used for LDF values in **c** to make it easier to distinguish differences within LAE groups.

**Thermal conduction at twist grain boundaries**. In Fig. 5, we compare LDFs and atomic thermal conductivities of three (001) twist GBs, viz. Σ41, Σ25 and Σ37, in order of increasing twist angle. In this case thermal conductivities are projected onto the GB planes, as opposed to parallel to the GB planes in the case of tilt GBs (Fig. 4). In the twist GBs, the LDFs are smaller than those of STGBs (as seen in Fig. 2a), but the structurally distorted sites are widely distributed about the GB plane, which is very different to the case of tilt GBs. In the case of the Σ41 twist GB, the dislocation lines, identified using the method of Stukowski et al.[43], are relatively far apart, and the LDF values are relatively low in the regions between them. These regions serve as thermal conduction highways, evidenced by the close match between regions of low LDF and high atomic thermal conductivity (Fig. 5a, b). In the case of the Σ37 GB, with its relatively high twist angle, all atoms on the GB plane are in distorted environments and the LDFs are uniformly high. The structural distortion thus correlates with low thermal conductivities across the GB plane in contrast to the rapid thermal conduction paths identified in the case of the Σ41 GB.

In contrast to the strong correlation between LDF and atomic thermal conductivity in the case of Σ41 and Σ37 twist GBs, the correlation in the case of the Σ25 twist GB is somewhat weaker. Even though its LDFs are lower than those of the Σ37 GB, especially in the inter-dislocation regions, their atomic thermal conductivities (and hence the effective thermal conductivity of the GB) are similar to those of the Σ37 GB. This difference indicates that the relationship between LDF and atomic thermal conductivity is non-linear; relatively small structural distortions to the lattice are sufficient to dampen the local thermal conduction strongly and thus very high LDFs may not be necessary to suppress thermal transport dramatically. This interpretation is consistent with the slow decrease in effective thermal conductivity exhibited by correlation II in Fig. 1a. Figure 5 also suggests that LDFs may be useful for identifying sites which induce strong phonon scattering and thus lower the effective thermal conductivity in the case of

twist GBs as well as for tilt GBs. Further discussion on the utility and limitations of the LDF is provided in Supplementary Note 4.

**Prediction models for thermal conductivity**. Motivated by the good correlation between LDF and atomic thermal conductivity described in the previous sections, we constructed a mathematical model for predicting thermal conductivities of GBs using multiple linear regression with $l_2$-norm (or ridge) regularisation. For this, we classified the LAEs into several groups according to the magnitude of their average LDF values by slicing the hierarchical clustering relationships in Fig. 3a in the manner described in Supplementary Fig. 5. We found that classifying the LAEs into six groups, viz., (1) bulk-like, (2) weakly strained, (3) moderately strained, (4) strongly strained, (5) moderately under-coordinated and (6) highly under-coordinated (as shown in Fig. 3a), is sufficient for accurate prediction of GB thermal conductivities. A summary of predictive performance using alternative numbers of LAE groups is also provided as Supplementary Fig. 5 and Supplementary Note 5.

Numbers of LAEs per unit area of a GB, $N_m$, for each LAE group ($m = 1$–6) were used as predictor variables, and fitting carried out using multiple linear regression (see the Methods section for details). As examples, Fig. 6a, b show the structures of $\Sigma5(310)/[001]$ and $\Sigma327(17\,\overline{19}\,2)/[111]$ STGBs, the Gaussian weighting function, $G(x)$, and plots of their $N_m$ values for each LAE group. These show that there are only highly distorted LAEs in the vicinity of the high-angle $\Sigma5(310)/[001]$ GB whereas there are both bulk-like and moderately distorted LAEs in the vicinity of the low-angle $\Sigma327(17\,\overline{19}\,2)/[111]$ GB.

The predictor model was trained using data from 70 randomly chosen symmetric GBs, and then validated using data from the remaining 22 GBs, including all four asymmetric tilt GBs. Figure 6c shows a parity plot of overall thermal conductivities calculated using perturbed MD against values predicted by the model. The root mean squared error (RMSE) and $R^2$ value are 1.28 Wm$^{-1}$K$^{-1}$ and 0.93, respectively, for the training data, and 1.30 Wm$^{-1}$K$^{-1}$ and 0.92, respectively, for the test data. These results demonstrate that GB thermal conductivity can be predicted with high precision from their local atomic structures alone, regardless of whether the GB is under standard or high pressure, a tilt, twist or twin GB. The prediction model also reliably estimated thermal conductivities of the asymmetric tilt GBs, confirming its good transferability as well as the efficacy of including a wide range of GB types in the training dataset. In addition, as seen in Fig. 6d, the regression coefficient is very high in the case of LAE group 1, where LDFs are very small (70.0 on average), i.e., the local environments are very similar to those in the crystal bulk, and very low for the other LAE groups decreasing gradually as LDF increases. These results again suggest that introducing GBs with relatively small structural distortions (e.g., low-angle GBs with dense GB cores) is an effective strategy for reducing thermal conductivity dramatically.

## Discussion

Similar to point defects such as vacancies, impurity atoms and interstitial atoms[44], GBs are known to limit phonon MFPs by causing diffuse scattering, and this is consistent with the results of our perturbed MD simulations. GBs can be thought of as extended planar defects or clusters of point defects, typically a few nanometres wide, so that deviations from the ideal lattice in the vicinity of GBs, as reflected in their LAEs and LDFs, are typically much larger than for isolated defects, making them able to scatter long-wavelength phonons much more effectively, resulting in much shorter MFPs in a polycrystal than in single crystal (in which MFPs are on the order of hundreds of nanometres or

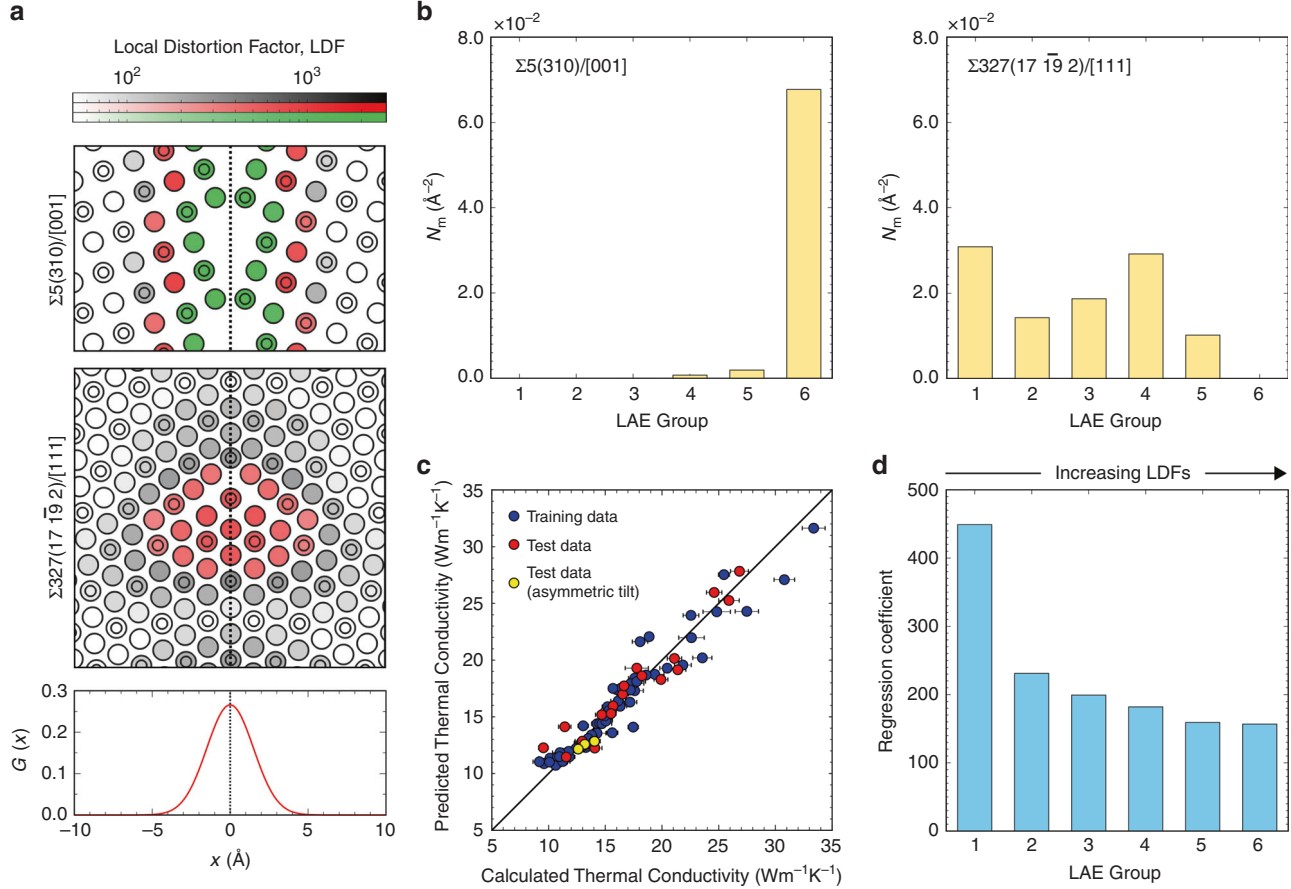

**Fig. 6 Regression modelling of thermal conductivity at GBs of MgO. a, b** Example of how predictor (input) variables $N_m$ were generated from GB structures for multiple linear regression. **a** LDFs in the vicinity of high-angle Σ5(310)/[001] and low-angle Σ327(17 $\overline{19}$ 2)/[111] STGBs, and the Gaussian function $G(x)$ centred on the GB plane used in calculating $N_m$. A log scale is used for LDF values to make it easier to distinguish differences within LAE groups. **b** Number of atoms per unit area in each LAE group, $N_m$, using hierarchical clustering results for the two GBs. **c** Parity plot of calculated against predicted GB thermal conductivities. Error bars indicate standard deviations in thermal conductivity calculated using perturbations of different magnitudes. **d** Ridge regression coefficients for $N_m$ of each LAE group. The higher the LAE group number, the larger the LDF values in the group.

several micrometres in the case of single crystal MgO[45]. Constructing an ML model with data from MD simulations of GBs shows that these effects can be predicted accurately from analysis of LAEs calculated with only a short cutoff (~4.5 Å).

The correlation between GB structure and thermal conductivity identified in this study should enable polycrystalline materials to be designed with more precisely controlled thermal conductivities, e.g., by identifying GBs with the desired microscopic behaviour for a given application and facilitating their formation in the material with appropriate synthesis methods and conditions. Although it is still very difficult to engineer GB structures directly at the atomic level, it is possible to increase the probability of their formation by tailoring grain orientation through thermal treatment, mechanical processing, use of substrates, and so on, as grains coming into contact within a narrower range of orientations are more likely to exhibit a particular GB structure with the desired LAEs. It should also be possible to examine the effect of dopants on GB thermal conductivities using this model, assuming suitable potential parameters are available for performing MD simulations, although the number of simulations required may increase substantially as a result of the increased degrees of freedom (dopant concentration, segregation sites and so on). Nevertheless, extending the ML method developed in this study to more complex crystal structures and compounds should enable a more comprehensive understanding of GB structure-property relationships to be obtained, so that the

next-generation of thermal materials can be designed more efficiently and effectively.

The method presented here, in which the relationship between thermal conductivity and local atomic distortions is identified through ML with a multidimensional dataset, can be readily applied to other structure-property relationships because of the universality of the SOAP descriptor, whether the cause of the distortion is point defects (isolated or clustered), dislocations, GBs, heterointerfaces or surfaces. When used in conjunction with a large dataset of defective structures such as those generated by atomistic materials modelling[46–48] using reliable interatomic potentials, quantification of complex structure-property relationships using ML techniques with SOAP-derived metrics has the potential to provide deeper insights into complex interface phenomena and greatly accelerate materials design of a broad range of technologically important materials. In some situations, however, it may be necessary to include directional information in the model so that properties more sensitive to anisotropy or that are highly directional can be predicted accurately. Methods for including directional information are discussed briefly in Supplementary Note 3 as a stimulus for future work.

In summary, we have used ML with data derived from the SOAP descriptor and perturbed MD to quantify the relationship between local atomic structure and overall thermal conductivity in standard- and high-pressure STGBs, twin, twist and asymmetric tilt GBs of MgO. The LDF, a simple metric based on the

SOAP descriptor, was found to correlate well with atomic thermal conductivity in a non-linear fashion. The prediction model constructed based on this insight revealed that even small structural distortions at GBs can reduce thermal conductivity dramatically, suggesting that the thermal conductivity of a polycrystalline material may be closely controlled by tailoring the number and distribution of such GBs through GB engineering. Although the importance of structural disorder at GBs has been posited by earlier researchers[20,49], to the best of our knowledge this is the first study to demonstrate quantitatively the correlation between structural distortion and suppression of thermal conductivity at the atomic level.

## Methods

**GB model construction**. Eighty-one standard-pressure STGBs of MgO constructed previously[25] were used together with an additional three (001) twist GBs, four [001] asymmetric tilt GBs, and four high-pressure STGBs generated using the method described previously[25,50]. Simulated annealing (SA) of initial structures was performed to obtain the stable atomic configurations of the GBs using equilibrium MD methods encoded in the Large-scale Atomic/Molecular Massively Parallel Simulator (LAMMPS) programme[51]. Initial configurations were constructed by tilting or twisting two half-crystals by a specific angle, and sandwiching an amorphous block of MgO between them. The amorphous block was obtained from a separate MD calculation by heating a perfect crystal of MgO to 8000 K. The rigid-ion Buckingham potential for MgO reported by Landuzzi et al.[52] was used in all cases.

SA simulations commenced with the GB model heated to 4000 K, and the temperature was decreased gradually to 50 K over 330 ps. This gradual cooling from high temperature allowed the atoms in the amorphous region to diffuse and find energetically favourable positions, so that a low-energy ordered GB structure was obtained for each initial configuration. The final atomic configuration for each GB model was obtained by performing geometry optimisation (at 0 GPa) using the General Utility Lattice Program (GULP)[53] on the structures obtained from SA simulations. In several cases, metastable GB structures (GBs with higher energies than the most stable form for that GB orientation at 0 GPa with atoms trapped in higher-energy local minima) were also obtained. These GB structures became lower in energy than the stable GB structures when geometry-optimised at higher pressures using GULP, so these were included as examples of high-pressure STGBs when developing the ML model.

We repeated the SA simulations 10 times for each symmetric GB and 50 times for each asymmetric GB using different initial velocity distributions to confirm that the most energetically stable atomic arrangement had been obtained. Structures of the Σ5 (310)/[001] GB were found to be in agreement with that determined using first-principles calculations[54], and a few dislocation core structures, which can be seen in low-angle STGBs with [001] and [1$\bar{1}$0] rotation axes, e.g., Σ41(540)/[001] and Σ51(1 1 10)/[1$\bar{1}$0] GBs, were found to be in excellent agreement with those observed by scanning transmission electron microscopy[25,55]. This gives us confidence that we successfully identified the lowest-energy (ground-state) structures. These GB models are available as Supplementary Data 1 in LAMMPS format. Two GB structure models are illustrated in Supplementary Fig. 6 as examples.

The excess volume per unit area of each GB, $\Delta V^{GB}$, was calculated using the following equation:

$$\Delta V^{GB} = \frac{V^{GB} - \frac{N^{GB}}{N^{SC}} V^{SC}}{2A} = \frac{V^{GB} - N^{GB}/\rho^{SC}}{2A} \quad (1)$$

where $V^{GB}$ and $V^{SC}$ are the volume of the GB model and unit cell, respectively, $N^{GB}$ and $N^{SC}$ are the number of atoms in the GB model and unit cell, respectively, and $\rho^{SC}$ is the number density of the unit cell.

**SOAP descriptor**. SOAP vectors of all atoms in MgO GBs were calculated using the Python-based software DScribe[56]. The SOAP descriptor is derived by fitting a set of spherical harmonics and radial basis functions to the 3-dimensional density distribution generated by placing Gaussian-smeared atomic densities on atoms within a specified cutoff radius about a central atom. The coefficients of the fit form a rotationally invariant power spectrum[57] which is compiled into a SOAP vector for that atom which contains all the information needed to reconstruct the LAE. Compiling SOAP vectors of atoms in the GB model into a matrix known as the local environment representation allows each particular GB structure to be described quantitatively and uniquely[38,41]. One of the advantages of the SOAP descriptor is that it also makes it possible to compare LAEs quantitatively, so that a dissimilarity (or, conversely, similarity) metric can be defined between two atoms[33] which varies smoothly with a change in neighbouring atom positions[38]. In this study, we used a non-normalised dissimilarity metric, $d$, defined as

$$d_{ij} = \sqrt{\mathbf{p}_i \cdot \mathbf{p}_i + \mathbf{p}_j \cdot \mathbf{p}_j - 2\mathbf{p}_i \cdot \mathbf{p}_j} \quad (2)$$

where $\mathbf{p}_i$ and $\mathbf{p}_j$ are the SOAP vectors of two atoms $i$ and $j$. If $\mathbf{p}_i$ and $\mathbf{p}_j$ are the SOAP vectors of a GB atom and its equivalent crystal bulk atom, the dissimilarity

metric represents how much the LAE of the GB atom differs from that of the bulk atom. We refer to this as the local distortion factor, LDF, defined as

$$LDF = \sqrt{\mathbf{p}_{GB} \cdot \mathbf{p}_{GB} + \mathbf{p}_{bulk} \cdot \mathbf{p}_{bulk} - 2\mathbf{p}_{GB} \cdot \mathbf{p}_{bulk}} \quad (3)$$

where $\mathbf{p}_{GB}$ and $\mathbf{p}_{bulk}$ are the SOAP vectors of a GB atom and an atom in the crystal bulk, respectively. A cutoff of 4.461 Å, corresponding to the average of the fourth and fifth nearest neighbour distances in MgO, was selected after preliminary testing of cutoffs both shorter and longer.

To compare LAEs and GB excess volume quantitatively, we defined the term total distortion factor, TDF, to be the sum of all LDFs at a GB normalised to the GB cross-sectional area, $A$,

$$TDF = \sum_i LDF_i / 2A \quad (4)$$

where $i$ is the index of an atom in the GB model. TDF is divided by two because each GB model produces two GBs under periodic boundary conditions. The calculated TDF and GB excess volume exhibited a linear relationship, especially in the case of high-angle tilt GBs formed under standard pressure (see Supplementary Fig. 7 and Supplementary Note 6). We also calculated the LDFs and TDFs using cutoffs of 3.313 and 3.923 Å, and confirmed that the relationship between TDF and excess volume was not overly sensitive to the choice of cutoff. Using large cutoff radii (~10 Å or greater) made it difficult to identify the GB core structure because it resulted in many more atoms being classified as having under-coordinated atoms in their spheres of influence.

The maximum degree of spherical harmonics, $l_{max}$, and the number of radial basis functions, $n_{max}$, were set to 9 and 12, respectively. In test calculations, it was found that the linear relationship between TDF and excess volume was insensitive to $l_{max}$ (even 0 produced similar results) but $n_{max}$ needed to be sufficiently large to achieve a good linear fit. We used spherical Gaussian type orbitals (as defined in Himanen et al.[56]) as radial basis functions, with a Gaussian width of 0.5 Å. Another implementation of the SOAP descriptor, the QUIP code[58], was also tested, and produced essentially the same linear relationship as DScribe (see Supplementary Fig. 8), indicating that the results reported here do not depend strongly on the particular implementation of the SOAP descriptor.

To extract a unique set of LAEs from each GB model in Fig. 2a, we performed complete-linkage clustering as implemented in Scipy[59] so that all combinations of atoms in each LAE group had $d$ values below a threshold value of 30.0. The threshold value was carefully chosen to maximise the performance of the prediction model without compromising interpretability of the classification groups. We also tested normalised forms of the SOAP vectors and other dissimilarity metrics such as the SOAP kernel and Gaussian kernel, but found that they make interpretation of the hierarchical clustering results difficult and reduce the predictive performance of the model. Further details are given in Supplementary Methods.

**Thermal conductivity calculations**. Overall thermal conductivities across the GB planes and grain interiors, which we refer to as effective thermal conductivities, were calculated using the perturbed MD method[60] for a few high-pressure tilt, twist and asymmetric tilt GB structures at 300 K. Custom-written code was added to LAMMPS for this purpose. In this method, lattice thermal conductivity in the $x$ direction is calculated according to

$$\kappa_{lattice} = \frac{1}{F_{ext} T} \lim_{t \to \infty} \langle J_x \rangle_t \quad (5)$$

where $F_{ext}$ is the magnitude of the perturbation, $T$ is the absolute temperature and $J_x$ is the heat flux in the $x$ direction. The microscopic heat flux is defined by Irving and Kirkwood[61] to be

$$\mathbf{J} = \sum_i \mathbf{J}_i = \sum_i \frac{1}{2V} \left[ \left\{ m_i \mathbf{v}_i^2 \mathbf{I} + \sum_j \phi_{ij} \mathbf{I} \right\} \mathbf{v}_i - \sum_j \left( \mathbf{F}_{ij} \cdot \mathbf{v}_i \right) \mathbf{r}_{ij} \right] \quad (6)$$

where $\mathbf{J}_i$ is the atomic contribution of atom $i$ to the heat flux, $V$ is the volume of the GB model (supercell), $m_i$ and $\mathbf{v}_i$ are the mass and velocity of atom $i$, respectively, $\phi_{ij}$ is the interatomic potential energy between atoms $i$ and $j$, $\mathbf{I}$ is a unit tensor of second rank and $\mathbf{F}_{ij}$ is the force exerted by atom $j$ on atom $i$. By substituting Eq. 6 into Eq. 5, atomic thermal conductivities $\kappa_i$, which are the atomic contributions to overall lattice thermal conductivity, can be calculated according to

$$\kappa_{lattice} = \sum_i \kappa_i = \sum_i \frac{1}{F_{ext} T} \lim_{t \to \infty} \langle J_{i,x} \rangle_t \quad (7)$$

where $J_{i,x}$ is the contribution of atom $i$ to the heat flux in the $x$ direction. As seen in Eq. 6, atomic thermal conductivities are proportional to the inverse of the supercell volume, and thus must be normalised by multiplying the supercell volume for comparison between GB models. In addition, the intensities in the thermal conductivity map in Figs. 4 and 5 also depend on the number of atoms in the depth direction, Gaussian-smeared atomic thermal conductivities projected onto the two-dimensional planes were divided by the cell depth. This procedure for calculating thermal conductivity is the same as reported in our previous work on STGBs[25]: For each GB orientation, models were constructed with three different half-crystal widths (distances between GB planes of as close to 4, 5 and 6 nm as feasible for that particular misorientation) by altering the number of bulk layers. MD simulations

were then performed in the NPT ensemble for 100 ps with a timestep of 1 fs for each model to determine its equilibrium cell dimensions at 300 K. Next, an NVT ensemble was applied for 100 ps with temperature scaling, followed by 300 ps using a Nosé-Hoover thermostat, to ensure thermal equilibrium had been reached. Perturbed MD simulations were then performed on the equilibrated GB models for 1.1 ns and the average heat flux of the last 1.0 ns used to calculate the thermal conductivity. The first 0.1 ns of data was discarded because this was the time needed for the system to transition from thermal equilibrium to a steady state under the perturbation. For each model, perturbed MD simulations were performed with at least four different magnitudes of the perturbation (after confirming the response was within the linear regime) and the average thermal conductivity calculated. The effective thermal conductivity for a width of exactly 5 nm was then extracted from a linear regression fit to these averaged thermal conductivities. Atomic thermal conductivities, plotted in Figs. 4 and 5, were extracted from the GB models with half-crystal widths of about 5 nm. Further details on the perturbed MD method are also available elsewhere[60,62–64].

**Machine learning**. LAEs identified for each GB model using the complete-linkage algorithm were grouped and classified using Ward's minimum variance method of hierarchical clustering[42] as implemented in SciPy[59], again using the dissimilarity metric $d$ in Eq. 2, as it is equivalent to the Euclidean distance. We also tested several other methods, such as the average method, but Ward's method was found to perform the most reliably and consistently. With this method, LAEs in the various GB structures were grouped into six different categories within three supergroups based on their level of lattice distortion.

The prediction model for thermal conductivity was constructed using the number of LAEs per unit area of a GB in each LAE group $m$, $N_m$, as input variables. Values of $N_m$ were weighted by a Gaussian function, $G$, of the distance, $x$, of the LAE's atom from the GB plane according to

$$N_m = \frac{1}{A}\sum_i^n G(x) = \frac{1}{A}\sum_i^n \frac{1}{\sqrt{2\pi\sigma^2}}\exp\left(-\frac{x^2}{2\sigma^2}\right) \quad (8)$$

where $A$ is the GB cross-sectional area, $n$ is the number of atoms in the LAE group, $i$ is the index of an atom in the LAE group, and $\sigma$ is the variance (set to 1.5 Å). $N_m$ corresponds to the number density of atoms in the vicinity of the GB plane decomposed into the contribution of each LAE group.

Fitting was performed using regularised multiple linear regression (Ridge regression) as implemented in scikit-learn[65]. Ridge regression shrinks the regression coefficients, $\boldsymbol{\beta}$, to prevent overfitting to the training data, by penalizing their size according to

$$\boldsymbol{\beta} = \underset{\boldsymbol{\beta}}{\mathrm{argmin}}\left\{\sum_i^t \left(y_i - \beta_0 - \sum_j^p x_{ij}\beta_j\right)^2 + \lambda\sum_j^p \beta_j^2\right\} \quad (9)$$

where $t$ is the number of training data, $y_i$ is the $i$th observed value, $p$ is the number of input variables, $x_{ij}$ is the $j$th component of the input variable for the $i$th training datum, $\beta_0$ and $\beta_j$ are the intercept and the $j$th regression coefficients, respectively, and $\lambda$ is the regularization parameter[66]. Because thermal conductivity should be zero when all $N_m$ are zero, i.e., there are no atoms in the vicinity of the GB plane, in this study the intercept $\beta_0$ was set to zero. For training data, 70 of the symmetric GB models were randomly selected with the proviso that each class of GB (namely, the six types of tilt GBs grouped by rotation axis, low-angle tilt GBs (open or dense), high-angle tilt GBs, twist GBs and high-pressure GBs) was represented at least once. The model was trained using $\lambda = 3 \times 10^{-4}$, determined through cross-validation. The remaining 18 symmetric GBs and all four asymmetric tilt GBs were used as test data to estimate the predictive performance. Input values $N_m$ were not standardised because this was found to reduce the predictive performance of the model.

## Data availability
GB models used in this study are available as Supplementary Data 1. Effective thermal conductivities of all the GB models used in multiple linear regression are summarised in Supplementary Tables 1 to 9. All other data that support the findings of this study are available from one of corresponding authors S.F. upon request.

## Code availability
Details of computer codes used in this study are provided in Supplementary Methods.

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

## Acknowledgements

This work was supported by "Materials Research by Information Integration" Initiative (MI²I) project of the Support Program for Starting Up Innovation Hub from the Japan Science and Technology Agency (JST) and Grant-in-Aid for Scientific Research on Innovative Areas 'New Materials Science on Nanoscale Structures and Functions of Crystal Defect Cores' from the Japan Society for the Promotion of Science (JSPS) [grant number 19H05786].

## Author contributions

S.F. conceived the research idea, carried out theoretical calculations, performed machine learning and wrote the paper. C.A.J.F and T.Y. contributed to writing of the paper with oversight by M.Y. T.Y. constructed grain boundary models. M.Y., C.A.J.F. and H.M. advised on the machine learning method and interpretation of results. All authors discussed the results, and read and commented on the paper.

## Competing interests

The authors declare no competing interests.
