## [Peer Review File · Nature Communications]

Reviewers' comments:

Reviewer #1 (Remarks to the Author):

The goal of the authors is to predict thermal conductivity across grain boundaries (GBs) using a descriptor of the GB structure. To do this, the authors propose using the SOAP descriptor to quantify the local atomic structure in the GB. The SAOP descriptor is then used to cluster the environments into five categories: (1) bulk-like, (2) weakly strained, (3) strongly strained, (4) slightly under-coordinated and (5) highly under-coordinated groups. The number of LAEs per unit area of a GB, were then used as the predictor variables. The authors use multiple linear regression with ridge regularization to train the thermal conductivity model.

Overall, this is excellent work showcasing the capabilities of structural descriptors to predict a complex interface property – thermal conductivity. However, the reviewer is concerned about the dataset chosen for this study. Without a larger dataset including mixed, general grain boundaries (beyond symmetric tilt and twist GBs), it is difficult to ascertain the robustness of this ML model. For publication in a high impact journal, it is necessary to go beyond a set of tilt and twist grain boundaries. I recommend the authors simulate more GBs with diverse crystallographic character and show that the ML model predicts the thermal conductivity accurately.

Reviewer #2 (Remarks to the Author):

NCOMMS-19-35132: Quantitative prediction of grain boundary thermal conductivities from local atomic environments

The manuscript uses SOAP-based metrics to characterize the atomic structure of grain boundaries and correlate the atomic structures with the thermal conductivity of the grain boundary. Once characterized with SOAP, the atomic structures are classified using a hierarchical clustering technique. The cluster groups are used for machine learning to predict the thermal conductivity of the grain boundaries. The different atomic structure metrics of local distortion factor, etc. are found to correlate with the atomic thermal conductivities.

The manuscript is well written and easy to follow. The literature review is good and supports the work. This is an excellent example of how machine learning and complex structural descriptors can provide insight into complicated problems like the one illustrated here of knowing how different GB structures give different properties. While the work is good, there are a number of issues that need to be addressed.

From a reproducibility standpoint, many details are missing, though this is a challenge for all computational work like this. While not necessary, the authors might consider posting their scripts/code as supplemental materials.

Line 136 states, "GBs whose misorientations are close to twin." Two comments here. One, I would highly recommend a table of all the GBs in the supplemental material. Then determine which each GB belongs too. Two, which are "close" to twin? What is the threshold for that? There are GBs that are Sigma3 GBs that are not twin like and GBs that are not Sigma 3 that have twin faceting. I would like to know what the threshold is for being close to a twin.

The subsection "Structure descriptor" starting on line 144 is mostly methods. I would suggest moving much of that to the methods section. You will likely still need some text here to state what the descriptors are, but you can define them in the methods.

In equation 1 you define a dissimilarity metric that gives values that appear to range from 0 to

~3000. In machine learning it is common to use a Kernel function, like a Gaussian kernel which will give similarity values between 0 and 1. While certainly not necessary, it could simplify your analysis in that all your numbers will now be between a specific range. That can help knowing there is an absolute maximum and then your LDF and LSF are different by one minus the other instead of having to calculate a d_{\max} value. Just something to consider.

LSF is defined using an equation in Equation 3, why not define LDF in an equation? I understand it is not needed, but seems inconsistent, and perhaps the better reason is that I looked for the mathematical definition of LDF early in the reading and couldn't easily find it. Just a suggestion though it is NOT necessary.

Figure 2, I am unclear whether all the points in a vertical column are all the LDF values for all the atoms in a GB (I think this is true) or the metastable configurations. Assuming I am correct and it is the former, this just needs to be labeled more clearly. If it is the latter, it opens a lot of questions, like were all the metastable states considered in the learning, etc. I am also unclear on another aspect of the Figure 2. In line 168 it states "each model classified by GB rotation axis in order of increasing misorientation angle." I don't see how this could be true for the 110 since according to the legend, you have low angle then high angle then low angle symbols. As you can see there is the potential for the reader to not get everything you hope out of this graph, I suggest making sure it is clear.

Line 179 states, "Comparison between the expanded perfect crystals and GB structures thus indicates that it is the change in coordination environment that has the largest influence on LAEs at GBs." This may be true, but once you've reduced any environment to an LDF value, you cannot tell whether it is high/low because of strain or coordination. Thus, while the classification lets you know the difference, LDF does not. I do agree the metric is valuable, but I think it is worth pointing out its limitations.

Small item, but line 210 talks about d values for 2 circled atoms. Should this be an LDF value or a d value? Why switch from so much discussion of LDF to d here?

Line 226 defines the LSF, and I find this to be slightly confusing. I think it just comes down to having large and small values and pointing out that LDF is small for bulk-like atoms and LSF is large for that same atom. I know this seems obvious but the wording seemed unnecessarily confusing. Going along with this same questioning, if the LDF can go up to 3000 why does the LSF range (which should be able to be as high as 120% of LDF) only goes up to 700.

I find it odd that you choose to talk about specific directions of atomic conductivities for some GBs and other directions for others. The tilt does parallel to the GB except when it doesn't work for one GB and then you talk about perpendicular. Then use perpendicular for twist GBs. It could be viewed as suspicious that you're only presenting the best data. When you present the conductivity of the GBs for the machine learning is that directional? Or isotropic? Can you not get a similar overall number at the atomic level?

Line 266 it states, "The structural distortion thus correlates with low thermal conductivities across the GB plane and the absence of rapid thermal conduction paths as were found in the case of the Sigma41 GB." I agree that this is a very logical conclusion. My question is whether anyone has asserted something like this before. Are you the first? If so, great and you might note that, if not, you may note that you confirm other results.

On line 274 it states, "relatively small structural distortions to the lattice are enough to dampen the local thermal conduction" and on line 277 it states, "confirms that LDFs can be reliably used to identify sites which induce strong phonon scattering and thus lower the effective thermal conductivity..." These two statements could seemingly be conflicting though they do not have to be. The relatively small distortions are listed as being non-linear but may even be non-monotonic,

meaning that it may be difficult to know when a distortion is sufficient to dampen the conductivity or not. If true, it is hard to know whether you can reliably identify the sites that lead to lower conductivity. Is there a threshold or some reliable value you know of? I agree that it has potential, but I don't know if the evidence confirms that it is reliable.

Line 318 states, "despite thermal conduction being the collective motion of atoms (i.e., phonons) whose MFP is orders of magnitude greater (on the order of hundreds of nanometres or several micrometres in the case of MgO." If thermal conduction really requires knowledge of atom motions on the micrometer scale, how can you reliably calculate atomic conductivity or even GB conductivity. I don't dispute that your statement is correct, but the way this is worded suggests that you'd need larger scale information than you have, and if that's the case how could you do what you did?

Line 367 states, "these GB structures were found to be the most stable at high hydrostatic pressures" I assume you're talking about the metastable structure that become the most stable. Please clarify which are found to be the most stable and how do you determine that they are more stable at high pressure, do you run the GULP minimization again at high pressure?

Line 370 states, "all confirmed to be reproducible" how did you confirm that they are reproducible? To what were the GBs compared? More detail needed here.

Line 379 talks about the SOAP descriptor. You don't have any way to distinguish between Mg and O atoms do you? Once turned into a SOAP vector, it only characterizes the position of the surrounding atoms right? Or does it use different Gauss distribution widths for the two different atoms? Whether you do or not should be stated in the manuscript.

Line 444 is the intercept you're referring to Beta_0?

Weird wording, line 40, "nanocrystalline materials with high grain boundary (GB) populations exhibit extremely low lattice thermal conductivity" Better wording here to fix two potential issues. One, I think they mean large population rather than high. Two, at first I understood this to mean nanocrystalline materials with large populations of GBs as opposed to nanocrystalline materials that do not have large populations. I believe they mean nanocrystalline materials, which have large populations of GBs.

Line 98 replace rigorousness with rigor

Line 149 fix this awkward sentence, "The coefficients of the fit form of a rotationally invariant power spectrum ..."

Reviewer #3 (Remarks to the Author):

This manuscript presents a computational study of the thermal conductivity of MgO grain boundaries (GBs) and shows how a convenient measure of local atomic structure (the LDF) correlates well with atomically resolved contributions to thermal conductivity. A machine learning approach is used to train a model to predict thermal conductivity based on structure alone and tests are presented that are encouraging. The results are very interesting and certainly novel. The results are also clearly presented and calculations appear to have been carried out correctly and carefully. I believe the findings and topic are suitable for publication in Nature Communications. However, there are some points that I would like the authors to address before I can recommend publication:

Comments:

1. It is suggested that access to thermal conductivity information through relatively low computational cost models could enable "more precise design of next-generation thermal materials as it allows GB structures exhibiting the desired thermal transport behaviour to be identified". I am less convinced by this since it is not common to engineer atomic structure of GBs in real materials. Instead one might need to identify a preferred material from a list of possibilities or consider how best to modify an existing material (e.g. by adding dopants) to obtain an optimal material for a given application. It is less clear whether this approach would be viable in such cases. Some comments in the manuscript on this point should be added.
2. Related to the point above the quality of the predictions of the model depend on having an interatomic potential that is capable of predicting both atomic structure and thermal conductivity. Even for a relatively simple material like MgO this is not always straightforward and for many technologically important materials there are no suitable potentials available.
3. Based on the information presented I found it difficult to assess how truly predictive and transferable the machine learning model is. Since the training and test GBs were selected at random (and not listed anywhere) it is not clear how similar or dissimilar they are. For example if one were to consider a new GB with a different character (e.g. different tilt axis) would one expect the model to perform well? Some comments on this as well as more details on the training and test GB sets should be added. Also it would be helpful to include all GB models as a openly accessible dataset.
4. On page 9: "463.8 for strongly strained, 170.0 for weakly strained" - it would be helpful to quantify these strains making reference to Fig. 2b.
5. As the authors note, thermal conductivity is directional whereas the LSF is not. This would appear to be a serious issue for many properties of interest which are directional and highly anisotropic, e.g. impurity diffusion. Could some comments be added on how this could be improved in the future?
6. One should clarify whether the rigid ion or shell model is used for MgO?

[Reply to all reviewers]

We are grateful to all reviewers for taking the time to read our manuscript. After receiving their constructive comments, we have made many improvements to the scheme and conditions used when applying machine learning (ML) to the prediction of grain boundary (GB) thermal conductivities. Before replying to the comments individually, we would like to outline the major changes in general terms.

1. Extension to More Diverse GBs

To ensure the transferability of the ML model to more general GBs, we constructed four asymmetric tilt GBs to include in the test dataset and calculated their effective thermal conductivities using the same methods as originally reported. These calculations confirmed that our ML model is capable of predicting thermal conductivities of complex (asymmetric tilt) GBs with as good accuracy as higher symmetry GBs. A description of the wide range of GBs used in our dataset has been added to the supplementary material, along with a list of all GBs used in this work.

2. Improved Grouping of LAEs

We modified the way of assigning atoms to local atomic environments (LAEs) in a GB model slightly to make the model more robust. In the original method, an atom was randomly assigned to a new LAE (this atom was used to represent this LAE), and the other atoms similar to this atom were considered to belong to the same LAE based on the dissimilarity metric d . The procedure was repeated until all atoms in a GB model were assigned to an LAE. With this former method, the set of LAEs in a GB model was affected by the random way of choosing atoms, and the atoms were sometimes not the best representation of the LAE set because its SOAP vector was not necessarily close to the centroid of the other atoms' SOAP vectors. To remedy this, we upgraded the model to use hierarchical clustering with a complete linkage algorithm for determining a unique LAE set for each GB model, whereby all combinations of atoms belonging to an LAE have d values lower than a threshold value. The atom with a SOAP vector closest to the centroid of the SOAP vectors of all atoms in the group was used to represent that LAE.

3. Improved Classification of LAE Groups

After performing many tests with the new ML model described above, we re-classified the LAEs into six groups labelled (1) bulk-like, (2) weakly strained, (3) moderately strained, (4) strongly strained, (5) moderately under-coordinated and (6) highly under-coordinated. This model preserves the straightforward link between LAEs and GB thermal conductivities of the earlier model, and the conclusions of our manuscript remain unchanged. Although it is possible to reduce the number of LAE groups slightly without compromising the predictive performance too much (for example, by

combining LAE groups (5) and (6), whose regression coefficients are almost the same), we did not do so because interpretation of the relationship between LAEs and GB thermal conductivities appears more intuitive using the above six groups.

We have also added more detailed discussions to the main manuscript and supplementary material regarding important points raised by the reviewers.

[List of changes]

1. Four asymmetric tilt GBs have been added to the test dataset in order to ensure the robustness and transferability of the constructed ML model.
2. The method of extracting a unique LAE set from a GB model has been modified to use complete-linkage clustering.
3. The number of LAE groups was changed to six, which we label (1) bulk-like, (2) weakly strained, (3) moderately strained, (4) strongly strained, (5) moderately under-coordinated and (6) highly under-coordinated groups.
4. To avoid confusion, use of the local similarity factor (LSF) has been removed from the manuscript. The relationship between LAEs and atomic thermal conductivities is simply discussed in terms of the local distortion factor (LDF).
5. All figures have been updated accordingly.
6. The text has been modified or extended in several places in response to comments by the reviewers (highlighted in the revised manuscript). The major changes are as follows:

Line 118-119 (referring to the complete list of GBs in Supplementary Information)

Lines 129-133 (an explanation of thermal conductivities of asymmetric tilt GBs)

Lines 171-177 (the relationship between strain, coordination number and local distortion factor)

Lines 192-197 (an explanation of LAE groups classified by hierarchical clustering)

Lines 208-212, 218-221 (explanations of the aims and implications of analyses using the d metric between GB atoms)

Lines 247-248 (the discussion of the anisotropy of thermal conductivities has been moved to Supplementary Information to avoid confusion)

Lines 315-317 (predictive performance for asymmetric tilt GBs)

Lines 335-350 (discussion of how to control thermal conductivity through engineering GB structures)

Lines 360-363 (discussion of how to apply the method developed to properties that are highly directional)

Lines 399-405 (discussion of the reproducibility of GB models)

Lines 432-435 (definition of the local distortion factor)

Lines 459-463 (the method for determining an LAE set of a GB model)

Lines 463-467 (reasons why the normalisation of SOAP vectors and other dissimilarity metrics were not used)

Lines 520-524 (how the test dataset were selected)

Lines 680-681, 690-693, 708-713 (addition of six references)

7. A complete list of GBs used in the study has been added to Supplementary Information.
8. An explanation of the different types of GBs in the dataset has been added to Supplementary Information.
9. A discussion on the lack of directionality in SOAP descriptor has been added to Supplementary Information.
10. A discussion on the potential of local distortion factors as a descriptor of GB thermal conductivity has been added to the Supplementary Information.
11. A more detailed explanation of the coding used to perform machine learning has been added to Supplementary Information.
12. Some mistakes have been corrected. The discussion and conclusions of the manuscript remain unchanged.
 - i. The values of N_m were corrected (the values had been mistakenly multiplied by $1/2\pi\sigma^2$ instead of $1/\sqrt{2\pi\sigma^2}$).
 - ii. The values of root mean squared error (RMSE) were corrected (the values of mean squared error (MSE) had been mistakenly used).
 - iii. Fig. 6c was corrected (the calculated and predicted thermal conductivities had been plotted inversely).
 - iv. Some typographical errors were corrected.

[Replies to comments by Reviewer #1]

The goal of the authors is to predict thermal conductivity across grain boundaries (GBs) using a descriptor of the GB structure. To do this, the authors propose using the SOAP descriptor to quantify the local atomic structure in the GB. The SOAP descriptor is then used to cluster the environments into five categories: (1) bulk-like, (2) weakly strained, (3) strongly strained, (4) slightly under-coordinated and (5) highly under-coordinated groups. The number of LAEs per unit area of a GB, were then used as the predictor variables. The authors use multiple linear regression with ridge regularization to train the thermal conductivity model. Overall, this is excellent work showcasing the capabilities of structural descriptors to predict a complex interface property – thermal conductivity. However, the reviewer is concerned about the dataset chosen for this study.

Comment #1-1: Without a larger dataset including mixed, general grain boundaries (beyond symmetric tilt and twist GBs), it is difficult to ascertain the robustness of this ML model. For publication in a high impact journal, it is necessary to go beyond a set of tilt and twist grain boundaries. I recommend the authors simulate more GBs with diverse crystallographic character and show that the ML model predicts the thermal conductivity accurately.

Reply #1-1: Thank you making this suggestion. To ascertain the robustness of the ML model, we calculated the thermal conductivities of four asymmetric GBs and found that they were predicted well with our model even though they were not used as training data in the multiple linear regression (Fig. 6c). This is because LAEs in the asymmetric tilt GBs also fall into the six general LAE groups identified using symmetric GB data in the original manuscript. We believe that this provides good evidence that our training GB data are sufficiently diverse to predict accurately the thermal conductivities of GBs with arbitrary orientations, i.e., more general GBs. In addition, although not mentioned in the original manuscript, two of the original symmetric tilt GBs described as $[2\bar{1}0]$ and $[3\bar{1}0]$ type boundaries, actually had two rotation axes; each was initially rotated about the $[001]$ axis before rotating about $[2\bar{1}0]$ and $[3\bar{1}0]$ axes, respectively, making them more complex than the other “single-rotation” GBs. All together the extended set of GBs exhibits a wide range of macroscopic properties (Σ value, GB energy and GB excess volume in Supplementary Information Fig. S2), meaning that the training dataset includes a wide range of GBs in terms of crystallography, energetic stability and local number density of atoms at the GB planes. As a result, local distortion factors (LDFs) of all LAEs in the GB models, which are a measure of the changes in coordination environment and bond strain, were also distributed over a wide range of values, as shown in Fig. S3.

Although it is possible that other mixed GBs contain some LAEs that differ from those included in our training data, we expect that they will still to fall into one of the six LAE groups we identified from hierarchical clustering. We tested this by constructing ML models with and without

the asymmetric GB data, and found that the six groupings using Ward's method were not much affected, and the predictive performance of the ML model remain unchanged, as seen by comparing the follow two parity plots.

Fig. R1 Parity plots of calculated against predicted GB thermal conductivities. **a** LAEs of asymmetric tilt GBs used as test data but not training data; **b** LAEs of asymmetric tilt GBs not included in training or test data.

This shows that finer discrimination between LAEs are unnecessary for predicting GB thermal conductivities in the case of the asymmetric boundaries, which have more complex LAEs than symmetric GBs. We are thus confident that our ML model is capable of predicting GB thermal conductivities of a diverse range of MgO GBs with sufficiently good accuracy.

A discussion of how the asymmetric tilt GBs were constructed has been added to the manuscript, and a summary of the wide range of GBs examined in this study has been added as Supplementary Information (Tables S1 to S9, Figs. S2 and S3, and associated notes).

[Replies to comments by Reviewer #2]

The manuscript uses SOAP-based metrics to characterize the atomic structure of grain boundaries and correlate the atomic structures with the thermal conductivity of the grain boundary. Once characterized with SOAP, the atomic structures are classified using a hierarchical clustering technique. The cluster groups are used for machine learning to predict the thermal conductivity of the grain boundaries. The different atomic structure metrics of local distortion factor, etc. are found to correlate with the atomic thermal conductivities.

The manuscript is well written and easy to follow. The literature review is good and supports the work. This is an excellent example of how machine learning and complex structural descriptors can provide insight into complicated problems like the one illustrated here of knowing how different GB structures give different properties. While the work is good, there are a number of issues that need to be addressed.

Comment #2-1: From a reproducibility standpoint, many details are missing, though this is a challenge for all computational work like this. While not necessary, the authors might consider posting their scripts/code as supplemental materials.

Reply #2-1: Almost all our analysis was carried out using Python libraries that are publicly available under open source licences, such as DScibe, SciPy and scikit-learn. To aid reproducibility, we have added a description of the key steps and Python commands used as Supplementary Note 2: The Machine Learning Code.

Comment #2-2: Line 136 states, "GBs whose misorientations are close to twin." Two comments here. One, I would highly recommend a table of all the GBs in the supplemental material. Then determine which each GB belongs too. Two, which are "close" to twin? What is the threshold for that? There are GBs that are Sigma3 GBs that are not twin like and GBs that are not Sigma 3 that have twin faceting. I would like to know what the threshold is for being close to a twin.

Reply #2-2: As suggested, we have added tables of all the GBs to Supplementary Information (Tables S1 to S9) grouped according to their GB type (rotation axis, etc.).

Our GB dataset contains two $\Sigma 3$ boundaries, namely $\Sigma 3(111)$ and $\Sigma 3(112)$ GBs, but only the $\Sigma 3(111)$ GB is a twin boundary. The $\Sigma 3(111)$ twin boundary is a special boundary with a high degree of symmetry in which atoms are fully coordinated even at the boundary plane (its GB energy and excess volume are very low as shown in Supplementary Information Table S2). We classified GBs as being "twin-like" if their misorientation angles are between 0° and 15° with respect to the

(111) plane, i.e., the $\Sigma 3(111)$ twin boundary. These GBs contain regions of LAEs that are similar to those in the twin boundary. In our GB dataset, four boundaries were identified as twin-like, namely $\Sigma 41(443)/[1\bar{1}0]$, $\Sigma 33(554)/[1\bar{1}0]$, $\Sigma 57(445)/[1\bar{1}0]$ and $\Sigma 295(\bar{1}\bar{5}\bar{1}\bar{3}14)/[112]$ GBs. The structures of the twin and twin-like boundaries are shown in Supplementary Information Fig. S1, coloured according to their coordination number and LDF values. $\Sigma 33(554)/[1\bar{1}0]$ and $\Sigma 295(\bar{1}\bar{5}\bar{1}\bar{3}14)/[112]$ GBs contain sections of coherently bonded, fully coordinated atoms at the boundaries like the twin. In contrast, $\Sigma 41(443)/[1\bar{1}0]$ and $\Sigma 57(445)/[1\bar{1}0]$ GBs do not contain such sections, but have another kind of symmetric structure in which the half-crystals have slid slightly along the boundary plane. Because of their similarity to the twin structure, these LAEs have lower LDF values compared with conventional high-angle GBs (Fig. 3b), resulting in their different thermal conduction behaviour compared to high- and low-angle GBs with no twin-like sections, as seen in Fig. 1a. We have added an explanation of this categorisation to the main text with further details in Supplementary Information as notes to Tables S2 and S4, with example structures in Figure S1.

(lines 139-141)

GBs with very high symmetry and thus high number density, viz., the $\Sigma 3(111)$ twin boundary and GBs with LAEs similar to those of it (labelled “twin-like” in Fig. 1a), appear to fall on a fourth correlation line, one flatter than correlations I or I’.

Comment #2-3: The subsection “Structure descriptor” starting on line 144 is mostly methods. I would suggest moving much of that to the methods section. You will likely still need some text here to state what the descriptors are, but you can define them in the methods.

Reply #2-3: We have moved the first part of this subsection to the methods section, retaining only those parts needed to explain the results of local distortion factor in the results section.

Comment #2-4: In equation 1 you define a dissimilarity metric that gives values that appear to range from 0 to ~3000. In machine learning it is common to use a Kernel function, like a Gaussian kernel which will give similarity values between 0 and 1. While certainly not necessary, it could simplify your analysis in that all your numbers will now be between a specific range. That can help knowing there is an absolute maximum and then your LDF and LSF are different by one minus the other instead of having to calculate a d_{max} value. Just something to consider.

Reply #2-4: In the original paper of Bartok et al. [R1], they used a normalised linear or polynomial kernel to define similarity to be between 0 and 1 (SOAP kernel), which can then be translated into dissimilarity values between 0 and 1 as suggested, but Rosenbrock et al. [R2] later reported that the normalised form has poorer “discriminative ability” of LAEs compared to the non-normalised form. We confirmed that this is also true for our case. The normalised SOAP kernel makes it difficult to interpret the LAE classifications and weakens the predictive performance of the model; the root mean squared error (RMSE) and R^2 value are 1.98 W/mK and 0.83, respectively, for training data and 2.71 W/mK and 0.66, respectively, for test data despite using 9 input variables. This is probably because the length of the SOAP vector, which correlates with the density of atoms around an atom site, is also important for classifying the LAEs.

After taking these points into consideration we decided it is better to use SOAP vectors and the SOAP kernel in non-normalised form as defined in Eq. (2). We have added an explanation of this to the Methods section as follows:

(lines 463-466)

We also tested normalised forms of the SOAP vectors and other dissimilarity metrics such as the SOAP kernel and Gaussian kernel, but found that they make interpretation of the hierarchical clustering results difficult and reduce the predictive performance of the model.

Comment #2-5: LSF is defined using an equation in Equation 3, why not define LDF in an equation? I understand it is not needed, but seems inconsistent, and perhaps the better reason is that I looked for the mathematical definition of LDF early in the reading and couldn't easily find it. Just a suggestion though it is NOT necessary.

Reply #2-5: Thank you for this suggestion. Although we removed use of LSF from the manuscript, as explained in the reply to comment #2-9, we have added a definition of LDF to the Methods section:

(lines 431-435)

We refer to this as the *local distortion factor*, LDF, defined as

$$\text{LDF} = \sqrt{\mathbf{p}_{\text{GB}} \cdot \mathbf{p}_{\text{GB}} + \mathbf{p}_{\text{bulk}} \cdot \mathbf{p}_{\text{bulk}} - 2\mathbf{p}_{\text{GB}} \cdot \mathbf{p}_{\text{bulk}}}, \quad (3)$$

where \mathbf{p}_{GB} and \mathbf{p}_{bulk} are the SOAP vectors of a GB atom and an atom in the crystal bulk, respectively.

Comment #2-6: Figure 2, I am unclear whether all the points in a vertical column are all the LDF values for all the atoms in a GB (I think this is true) or the metastable configurations. Assuming I am correct and it is the former, this just needs to be labeled more clearly. If it is the latter, it opens a lot of questions, like were all the metastable states considered in the learning, etc. I am also unclear on another aspect of the Figure 2. In line 168 it states “each model classified by GB rotation axis in order of increasing misorientation angle.” I don’t see how this could be true for the 110 since according to the legend, you have low angle then high angle then low angle symbols. As you can see there is the potential for the reader to not get everything you hope out of this graph, I suggest making sure it is clear.

Reply #2-6: Figure 2 indeed shows all the LDF values for all the LAEs (atoms) in a GB as a vertical column. To make this clearer, we added vertical lines between columns for different GBs with an explanation in the figure caption.

Regarding misorientation angle, we apologise for this confusion and our poor explanation. The term “misorientation angle” should be “tilt or twist” angle. The designation of “low-angle” and “high-angle” is taken relative to the single crystal; because of the high symmetry of the rock-salt structure, for certain axes rotation about a high angle returns the structure to a single crystal. For example, for the [001] axis, rotations of 0° and 90° produce a single crystal with (100) and (110) planes on the boundary, so GBs whose tilt angles are close to these angles are low-angle boundaries with respect to (100) and (110) planes, respectively (misorientations relative to the single crystal are low in both cases but the tilt angle is high for (110) low-angle boundaries). In the case of the [110] axis, the critical angle is 180° . To correct this, we have changed the term “misorientation angle” to “tilt or twist angle”, and added the above information as notes to Supplementary Tables S1 and S2.

Comment #2-7: Line 179 states, “Comparison between the expanded perfect crystals and GB structures thus indicates that it is the change in coordination environment that has the largest influence on LAEs at GBs.” This may be true, but once you’ve reduced any environment to an LDF value, you cannot tell whether it is high/low because of strain or coordination. Thus, while the classification lets you know the difference, LDF does not. I do agree the metric is valuable, but I think it is worth pointing out its limitations.

Reply #2-7: Thank you for pointing this error. It is true that LDFs do not themselves indicate whether strain or coordination is more important, and after re-examining the effect of strain and coordination deficiency on LAEs in terms of averages and standard deviations of LDFs for fully coordinated and under-coordinated atoms we found that both factors affect LDFs strongly, contrary

to our original statement. We have corrected the text as follows:

(lines 171-177)

Although LDFs by themselves do not indicate whether strain or coordination environment has the stronger effect, separate analysis showed that both of them are important, with contributions of similar magnitude in many cases. For example, the average and standard deviation of LDFs of atoms with first-nearest neighbour coordination deficiencies of 0, 1 and 2 are 456.6 ± 392.4 , 1270.0 ± 608.4 and 1483.5 ± 580.0 , respectively. The LDF values increase with increasing under-coordination but also have high standard deviations because of large variations in bond strain about atoms with different LAEs.

Comment #2-8: Small item, but line 210 talks about d values for 2 circled atoms. Should this be an LDF value or a d value? Why switch from so much discussion of LDF to d here?

Reply #2-8: This should be a d value in this case. LDFs are a special case of the d metric, specifically comparing local distortion in the atomic environment of a GB atom with an atom in the crystalline bulk. In this part of the discussion we compare LAEs between different types of GBs, and thus calculate d values for the different GB atom LAEs. As the LDF metric is non-linear, comparing GBs using their LDFs is not as informative as using the direct d values, as the distinctions between them are “drowned out” by the larger difference with the crystal bulk atoms. Using d values directly allows better discrimination between LAEs of GBs, at the same time showing that the atomic environments are not so very different between various types of GBs, in order to confirm that our training dataset is sufficiently broad and diverse to apply to all conceivable (i.e., general) GBs. This provides confidence in the transferability of the model because other GBs are composed of atoms with LAEs whose magnitudes are encompassed by or similar to those in our fitted model. We have modified the text to explain this better as follows:

(lines 208-212)

LDF values quantify the local distortion relative to the ideal crystal bulk, but do not directly measure differences in LAEs between GBs. To better assess the range of LAEs exhibited by different types of GBs, we thus also calculated d values between all atoms in one GB model with those in another. This revealed that similar LAEs frequently occur in other GBs, with greater differences occurring for high-pressure and high-angle STGBs than for others.

(lines 218-221)

In other words, the range of LAEs provided by a sufficiently large and diverse sample of GB structures (92 in our case) is expected to encompass those encountered in GBs with other misorientations, higher complexity or lower symmetry.

Comment #2-9: Line 226 defines the LSF, and I find this to be slightly confusing. I think it just comes down to having large and small values and pointing out that LDF is small for bulk-like atoms and LSF is large for that same atom. I know this seems obvious but the wording seemed unnecessarily confusing. Going along with this same questioning, if the LDF can go up to 3000 why does the LSF range (which should be able to be as high as 120% of LDF) only goes up to 700.

Reply #2-9: There are a couple of reasons why the LSF range only goes up to 700: (1) d_{\max} is not the maximum of LDFs shown in Fig. 2 but the maximum of the LDFs of the GBs being compared, i.e., standard- and high-pressure $\Sigma 25(710)/[001]$ and $\Sigma 5(310)/[001]$ STGBs in Fig. 4 and three twist GBs in Fig. 5. (2) The LSF values are Gaussian-smeared. We did this only for ease of comparison between LAEs and atomic thermal conductivities, but we agree that they are unnecessarily confusing and do not add much. Thus, in the newest version of the manuscript, we have decided not to use LSF but compare the LDF values and atomic thermal conductivities directly in Figs. 4 and 5.

Comment #2-10: I find it odd that you choose to talk about specific directions of atomic conductivities for some GBs and other directions for others. The tilt does parallel to the GB expect when it doesn't work for one GB and then you talk about perpendicular. Then use perpendicular for twist GBs. It could be viewed as suspicious that you're only presenting the best data. When you present the conductivity of the GBs for the machine learning is that directional? Or isotropic? Can you not get a similar overall number at the atomic level?

Reply #2-10: All GB thermal conductivities reported are perpendicular to the GB plane because this is the direction in which thermal energy is transported from grain to grain. The discussion about thermal conduction parallel to the $\Sigma 5(310)/[001]$ boundary was included to explain the exception to the correlation between LDF values and atomic thermal conductivities for this GB, i.e., why atoms at the boundary with high LDF values exhibited the highest atomic thermal conductivities, whereas in the case of other GBs LDF values overall correlate well with atomic thermal conductivities. In short, the anisotropy in bond distances in the $\Sigma 5(310)/[001]$ GB is much larger than for other GBs, so that the LDF, which is non-directional, is large, even though perpendicular to the GB the distortion is

small (and hence thermal conductivity high). We apologise for explaining this poorly in the original manuscript, and have attempted to give a clearer explanation as follows:

(lines 237-250)

These plots reveal that, overall, there is strong negative correlation between LDF and atomic thermal conductivity in these two cases. One exception to this is the standard-pressure $\Sigma 5(310)/[001]$ STGB, in which LDFs of the innermost atoms (Fig. 4b) are high and their atomic thermal conductivities (Fig. 4d) are the highest of all atoms in the GB structure. This inversion of the correlation is because the SOAP vector, and hence LDF, are non-directional, whereas there is a large anisotropy in the bond distances and hence components of atomic thermal conductivity of the $\Sigma 5(310)/[001]$ GB, with single pairs of atoms across the GB plane acting like thermal conduction bottlenecks. Distances between atoms perpendicular to the GB plane are similar to those in the bulk, but much longer parallel to it in the $[\bar{1}\bar{3}0]$ direction, resulting in a large LDF factor (maps of the components of atomic thermal conductivity perpendicular and parallel to the GB plane are compared in Supplementary Fig. S4). Such bottlenecks generally only occur in high-angle STGBs, but in low densities dispersed between low-conductivity voids, so their effect on the overall thermal conductivity is small.

Comment #2-11: Line 266 it states, “The structural distortion thus correlates with low thermal conductivities across the GB plane and the absence of rapid thermal conduction paths as were found in the case of the Sigma41 GB.” I agree that this is a very logical conclusion. My question is whether anyone has asserted something like this before. Are you the first? If so, great and you might note that, if not, you may note that you confirm other results.

Reply #2-11: To the best of our knowledge, this is the first time that the impact of structural distortion on the GB thermal conductivity has been studied quantitatively at the atomic level, although some earlier studies (refs. R3 and R4, for example) showed that structural disorder at GBs is important for suppressing thermal conductivity. We have added these points to the end of the *Discussion* section as follows:

(lines 372-375)

Although the importance of structural disorder at GBs has been posited by earlier researchers [20,48], to the best of our knowledge this is the first study to demonstrate quantitatively the correlation between structural distortion and suppression of thermal conductivity at the atomic level.

Comment #2-12: On line 274 it states, “relatively small structural distortions to the lattice are enough to dampen the local thermal conduction” and on line 277 it states, “confirms that LDFs can be reliably used to identify sites which induce strong phonon scattering and thus lower the effective thermal conductivity...” These two statements could seemingly be conflicting though they do not have to be. The relatively small distortions are listed as being non-linear but may even be non-monotonic, meaning that it may be difficult to know when a distortion is sufficient to dampen the conductivity or not. If true, it is hard to know whether you can reliably identify the sites that lead to lower conductivity. Is there a threshold or some reliable value you know of? I agree that it has potential, but I don’t know if the evidence confirms that it is reliable.

Reply #2-12: We agree that the statement “confirms that LDFs can be reliably used to identify sites which induce strong phonon scattering and thus lower the effective thermal conductivity” was putting the case too strongly. Plots of LDF densities for each LAE group (added as Supplementary Fig. S3) show that there is overlap between the groups. This is particularly the case at smaller LDFs where there is substantial overlap between “bulk-like” and “weakly strained” groups, and in this case it is possible the atomic thermal conductivity to LDF relationship is non-monotonic (for example, as a result of an atom being in compression). To see what effect removing this overlap has, we constructed ML models using LAE groups differentiated by LDF thresholds alone. These models exhibited moderately good predictive performance but the regression coefficients of the LAE groups fluctuated strongly depending on threshold values selected, making it difficult to interpret the thermal suppression mechanism consistently. This is mainly because separating atoms into LAE groups by LDF alone does not distinguish between the type of atom environment as reliably as with Ward’s hierarchical clustering method. The relation between LDF and thermal conductivity appears to be clearer for highly distorted LAE groups, as their regression coefficients decrease steadily. In terms of a threshold value, LDFs > 300 appear to correspond to large decreases in thermal conductivity. Thus, while LDFs can be a useful general guide, they need to be considered in groups of atoms, e.g., visually using plots such as in Fig. 5 for reliable interpretation. We have modified the discussion about the potential of using LDF to identify conduction-suppressing sites in the main text, and added some more detailed notes as Supplementary Note 1: Local Distortion Factors.

(lines 283-288)

This interpretation is consistent with the slow decrease in effective thermal conductivity exhibited by correlation II in Fig. 1. Fig. 5 also suggests that LDFs may be useful for identifying sites which induce strong phonon scattering and thus lower the effective thermal conductivity in the case of twist GBs as well as for tilt GBs. Further discussion on the utility and limitations of the LDF is provided in Supplementary Note 1.

Comment #2-13: Line 318 states, “despite thermal conduction being the collective motion of atoms (i.e., phonons) whose MFP is orders of magnitude greater (on the order of hundreds of nanometres or several micrometres in the case of MgO.” If thermal conduction really requires knowledge of atom motions on the micrometer scale, how can you reliably calculate atomic conductivity or even GB conductivity. I don’t dispute that your statement is correct, but the way this is worded suggests that you’d need larger scale information than you have, and if that’s the case how could you do what you did?

Reply #2-13: MFPs on the order of hundreds of nanometres or several micrometres can be examined using molecular dynamics because of the periodic boundary conditions. Note that the phonon wavelength and MFP are different, and thermal conductivity of MgO can be reproduced well if the supercell size is larger than the wavelengths of phonons that contribute to its thermal conduction. To investigate the effect of finite supercell size, thermal conductivities of monocrystalline MgO was calculated using cubic cells with sides of 2.1, 2.5, 3.4, and 3.8 nm, respectively. We found that thermal conductivity was well converged with a supercell size of 2.1 nm. Because we used large supercells with sides of 2.1 nm or longer, we could reliably calculate atomic thermal conductivities and GB thermal conductivities using molecular dynamics (within the limits of the empirical potential model). Please also note that when we stated that MFPs are on the order of hundreds of nanometres or several micrometres, this is referring to a single crystal. In our GB models, the MFPs of phonons are much smaller because of scattering. The main point of the sentence in question was that using a short cutoff of 4.5 Å in the SOAP descriptor is sufficient for constructing a good ML prediction model, indicating that a short-range disturbance to the lattice is sufficient to scatter phonons, even though in single crystals the MFPs are much larger. We have re-written the paragraph to be clearer as follows:

(lines 324-334)

Similar to point defects such as vacancies, impurity atoms and interstitial atoms [43], GBs are known to limit phonon MFPs by causing diffuse scattering, and this is consistent with the results of our perturbed MD simulations. GBs can be thought of as extended planar defects or clusters of point defects, typically a few nanometres wide, so that deviations from the ideal lattice in the vicinity of GBs, as reflected in their LAEs and LDFs, are typically much larger than for isolated defects, making them able to scatter long-wavelength phonons much more effectively, resulting in much shorter MFPs in a polycrystal than in single crystal (in which MFPs are on the order of hundreds of nanometres or several micrometres in the case of single crystal MgO [44]). Constructing an ML model with data from MD simulations of GBs shows

that these effects can be predicted accurately from analysis of LAEs calculated with only a short cutoff ($\sim 4.5 \text{ \AA}$).

Comment #2-14: Line 367 states, “these GB structures were found to be the most stable at high hydrostatic pressures” I assume you’re talking about the metastable structure that become the most stable. Please clarify which are found to be the most stable and how do you determine that they are more stable at high pressure, do you run the GULP minimization again at high pressure?

Reply #2-14: Yes, we performed static energy minimization calculations using GULP at high hydrostatic pressures for both the most stable and metastable GBs obtained under 0 GPa, and found that metastable GBs at 0 GPa become the most stable at the high pressures, and vice versa. Those GBs labelled "high-pressure" are metastable under 0 GPa, whereas all others are the most stable configuration for their particular misorientation. We amended the text to read as follows:

(lines 394-398)

In several cases, metastable GB structures (GBs with higher energies than the most stable form for that GB orientation at 0 GPa with atoms trapped in higher-energy local minima) were also obtained. These GB structures became lower in energy than the stable GB structures when geometry-optimised at higher pressures using GULP, so these were included as examples of high-pressure STGBs when developing the ML model.

Comment #2-15: Line 370 states, “all confirmed to be reproducible” how did you confirm that they are reproducible? To what were the GBs compared? More detail needed here.

Reply #2-15: Where possible we compared the GB structures obtained from our simulations with those observed by TEM or obtained from first-principles calculations for the same GB plane and misorientation. For the remaining GBs, whose structures have not been reported in the literature, we repeated the SA simulations 10 times for each symmetric GB and 50 times for each asymmetric GB to confirm that the same (stable or metastable) structure was obtained. To make this clearer we modified the text as follows:

(lines 399-405)

We repeated the SA simulations 10 times for each symmetric GB and 50 times for each asymmetric GB using different initial velocity distributions to confirm that the most

energetically stable atomic arrangement had been obtained. Structures of the $\Sigma 5(310)/[001]$ GB were found to be in agreement with that determined using first-principles calculations [53], and a few dislocation core structures], which can be seen in low-angle STGBs with $[001]$ and $[1\bar{1}0]$ rotation axes, e.g., $\Sigma 41(540)/[001]$ and $\Sigma 51(1\ 1\ 10)/[1\bar{1}0]$ GBs, were found to be in excellent agreement with those observed by scanning transmission electron microscopy [25,54].

Comment #2-16: Line 379 talks about the SOAP descriptor. You don't have any way to distinguish between Mg and O atoms do you? Once turned into a SOAP vector, it only characterizes the position of the surrounding atoms right? Or does it use different Gauss distribution widths for the two different atoms? Whether you do or not should be stated in the manuscript.

Reply #2-16: No, we have no way of distinguishing SOAP vectors of Mg and O atoms. In order to treat all SOAP vectors equivalently, we specified that the components be listed in the same order as the coordination shells, i.e., in SOAP vectors centred on Mg atoms, O-O interactions appear first, then those Mg-O interactions, the Mg-Mg interactions, whereas for those of O atoms, Mg-Mg terms are listed first, then Mg-O, then O-O, rather than in order of increasing atomic number, which is the default setting in the Dscribe code. The Gauss distribution widths we used were 0.5 Å for both Mg and O atoms. We have added a comment describing this to Supplementary Note 2.

Comment #2-17: Line 444 is the intercept you're referring to Beta_0?

Reply #2-17: Yes. We have changed the text to "the intercept β_0 ".

Comment #2-18: Weird wording, line 40, "nanocrystalline materials with high grain boundary (GB) populations exhibit extremely low lattice thermal conductivity" Better wording here to fix two potential issues. One, I think they mean large population rather than high. Two, at first I understood this to mean nanocrystalline materials with large populations of GBs as opposed to nanocrystalline materials that do not have large populations. I believe they mean nanocrystalline materials, which have large populations of GBs.

Reply #2-18: The reviewer is correct. We modified the text as follows:

(lines 34-37)

Recent studies using advanced nanostructuring techniques have shown that nanocrystalline materials, which have large grain boundary (GB) populations, exhibit extremely low lattice thermal conductivities [1,5,10,11], even when the bulk form is thermally conductive, e.g., elemental silicon [12,13].

Comment #2-19: Line 98 replace rigorousness with rigor

Reply #2-19: We replaced "rigorousness" with "rigour".

Comment #2-20: Line 149 fix this awkward sentence, "The coefficients of the fit form of a rotationally invariant power spectrum ..."

Reply #2-20: We modified the sentence as follows and moved it to the Methods section:

(lines 419-420)

The coefficients of the fit form a rotationally invariant power spectrum [56] which is compiled into a SOAP vector for that atom.

[Replies to comments by Reviewer #3]

This manuscript presents a computational study of the thermal conductivity of MgO grain boundaries (GBs) and shows how a convenient measure of local atomic structure (the LDF) correlates well with atomically resolved contributions to thermal conductivity. A machine learning approach is used to train a model to predict thermal conductivity based on structure alone and tests are presented that are encouraging. The results are very interesting and certainly novel. The results are also clearly presented and calculations appear to have been carried out correctly and carefully. I believe the findings and topic are suitable for publication in Nature Communications. However, there are some points that I would like the authors to address before I can recommend publication:

Comment #3-1: It is suggested that access to thermal conductivity information through relatively low computational cost models could enable "more precise design of next-generation thermal materials as it allows GB structures exhibiting the desired thermal transport behaviour to be identified". I am less convinced by this since it is not common to engineer atomic structure of GBs in real materials. Instead one might need to identify a preferred material from a list of possibilities or consider how best to modify an existing material (e.g. by adding dopants) to obtain an optimal material for a given application. It is less clear whether this approach would be viable in such cases. Some comments in the manuscript on this point should be added.

Reply #3-1: We agree that engineering atomic structures of GBs in practical materials is still some way off in the future, whereas with current technology it is only possible to "encourage" grains to orientate in a particular direction by controlling grain size and shape (e.g., texturing), or to modify the grain boundary chemistry through doping with atoms that segregate, so as to "tune" the thermal conductivity. The influence of GBs on thermal conductivity thus becomes a matter of statistics (the numbers, distributions and interconnectivity of GBs). Our statement about identifying GB structures with the desired behaviour was made with this point in mind; rather than engineering those GBs directly or explicitly, at this stage we envisage using the information gained from our ML model to select processing conditions that have a greater probability of forming GBs (or a component of that GB) with a particular desired misorientation (or close to it).

Concerning the effect of dopants, although we are yet to confirm it, in principle our method can be used to quantify the effect of dopants (either singularly or in combination) on thermal conductivity (assuming suitable and reliable potential parameters are available for a particular doping elements in a given host material). This, however, will require a much larger number of MD simulations to be performed because of the added degrees of freedom e.g., dopant site and concentration. Prompted by the reviewer's comments we have explored these ideas briefly in the text

as follows:

(lines 335-350)

The correlation between GB structure and thermal conductivity identified in this study should enable polycrystalline materials to be designed with more precisely controlled thermal conductivities, e.g., by identifying GBs with the desired microscopic behaviour for a given application and facilitating their formation in the material with appropriate synthesis methods and conditions. Although it is still very difficult to engineer GB structures directly at the atomic level, it is possible to increase the probability of their formation by tailoring grain orientation through thermal treatment, mechanical processing, use of substrates, and so on, as grains coming into contact within a narrower range of orientations are more likely to exhibit a particular GB structure with the desired LAEs. It should also be possible to examine the effect of dopants on GB thermal conductivities using this model, assuming suitable potential parameters are available for performing MD simulations, although the number of simulations required may increase substantially as a result of the increased degrees of freedom (dopant concentration, segregation sites, and so on). Nevertheless, extending the ML method developed in this study to more complex crystal structures and compounds should enable a more comprehensive understanding of GB structure-property relationships to be obtained, so that the next-generation of thermal materials can be designed more efficiently and effectively.

Comment #3-2: Related to the point above the quality of the predictions of the model depend on having an interatomic potential that is capable of predicting both atomic structure and thermal conductivity. Even for a relatively simple material like MgO this is not always straightforward and for many technologically important materials there are no suitable potentials available.

Reply #3-2: This is an important point, and one of the reasons why for this study we looked at probably the simplest oxide, MgO, for which the classical interatomic potential is considered robust, as a “test of concept”. Our method, of course, is not limited to Buckingham potentials, but can treat any material describable using potential forms, e.g., those available in the LAMMPS code. It is true, however, many technologically important materials are currently too complex (multiphase or composite systems, nanoporous materials, or those with elements particularly difficult to describe using conventional force-fields (e.g., Pb, Sb)). In these cases, alternative simulation methods need to be used, e.g., order-n first-principles calculations and ML interatomic potentials. As computing power continues to improve, we believe some of these difficulties will be overcome with time, and the variety of materials that can be treated will increase accordingly. To address this concern, we

have added the following to the Discussion section:

(lines 355-360)

When used in conjunction with a large dataset of defective structures such as those generated by atomistic materials modelling [45-47] and accurate interatomic potentials, quantification of complex structure-property relationships using ML techniques with the SOAP-derived metrics has the potential to greatly accelerate materials design of a broad range of technologically important materials.

Comment #3-3: Based on the information presented I found it difficult to assess how truly predictive and transferable the machine learning model is. Since the training and test GBs were selected at random (and not listed anywhere) it is not clear how similar or dissimilar they are. For example if one were to consider a new GB with a different character (e.g. different tilt axis) would one expect the model to perform well? Some comments on this as well as more details on the training and test GB sets should be added. Also it would be helpful to include all GB models as a openly accessible dataset.

Reply #3-3: We constructed a number of additional, more complex (i.e., asymmetric) GBs and performed further tests to confirm that the ML model is accurate and transferable. These data were not used in the training of the ML model, but their thermal conductivities were predicted well (Fig. 6c has been updated to show this), giving us increased confidence in the robustness of our model.

As requested by all reviewers, we have added tables listing all the GBs used in this study to Supplementary Information (Tables S1 to S9). These show that the dataset contains a wide variety of GB types (in terms of coincident site densities, $1/\Sigma$, GB energies and GB excess volumes in Supplementary Figs. S2 and S3) with diverse LAEs. We have updated and improved the text and figures as appropriate. We will make all the GB models available to the public as Supplementary Files in LAMMPS format.

Comment #3-4: On page 9: “463.8 for strongly strained, 170.0 for weakly strained” - it would be helpful to quantify these strains making reference to Fig. 2b.

Reply #3-4: We have added the following sentence about the strain levels, making reference to Fig. 2b.

(lines 192-194)

For reference, from Fig. 2b the LDFs of weakly strained, moderately strained and strongly strained groups correspond to average bond elongations of roughly 0.4%, 0.8% and 1.6%, respectively.

Comment #3-5 As the authors note, thermal conductivity is directional whereas the LSF is not. This would appear to be a serious issue for many properties of interest which are directional and highly anisotropic, e.g. impurity diffusion. Could some comments be added on how this could be improved in the future?

Reply #3-5: Yes, this is an important issue which is not straightforward to solve, but something we are keenly aware of and have been giving considerable thought to, as GBs with highly anisotropic interatomic bond lengths may be problematic. One possible solution is to include an additional metric describing the connectivity (directions) between atoms (and hence their LDFs). Another possibility is to replace the SOAP descriptor with one that includes directionality implicitly, such as “force-field-inspired” descriptors proposed by Choudhary and coworkers [R5]. We have added a brief discussion of this issue as Supplementary Note for Fig. S4.

(lines 360-363)

In some situations, however, it may be necessary to include directional information in the model so that properties more sensitive to anisotropy or that are highly directional can be predicted accurately. Methods for including directional information are discussed briefly in Supplementary Information as a stimulus for future work.

Comment #3-6: One should clarify whether the rigid ion or shell model is used for MgO?

Reply #3-6: We used a rigid ion model for MgO, and modified the text as follows:

(lines 386-387)

The rigid-ion Buckingham potential for MgO reported by Landuzzi et al. [51] was used in all cases.

References

- [R1] Bartók, A. P., Kondor, R. & Csányi, G. On representing chemical environments. *Phys. Rev. B* **87**, 1–16 (2013).
- [R2] Rosenbrock, C. W., Homer, E. R., Csányi, G. & Hart, G. L. W. Discovering the building blocks of atomic systems using machine learning: Application to grain boundaries. *npj Comput. Mater.* **3**, 1–7 (2017).
- [R3] Schelling, P. K., Phillpot, S. R. & Keblinski, P. Kapitza conductance and phonon scattering at grain boundaries by simulation. *J. Appl. Phys.* **95**, 6082–6091 (2004).
- [R4] Spiteri, D., Anaya, J. & Kuball, M. The effects of grain size and grain boundary characteristics on the thermal conductivity of nanocrystalline diamond. *J. Appl. Phys.* **119**, 085102 (2016).
- [R5] Choudhary, K., DeCost, B. & Tavazza, F. Machine learning with force-field-inspired descriptors for materials: Fast screening and mapping energy landscape. *Phys. Rev. Materials* **2**, 083801 (2018).

REVIEWERS' COMMENTS:

Reviewer #1 (Remarks to the Author):

I am happy with the changes made by the authors.

Reviewer #3 (Remarks to the Author):

The authors have addressed my comments and I recommend it is accepted for publication.

Keith McKenna